# Climatology of aerosol pH and its controlling factors at the Melpitz continental background site in central Europe

Vikram Pratap[1], Christopher J. Hennigan[1], Bastian Stieger[2,3], Andreas Tilgner[2], Laurent Poulain[2], Dominik van Pinxteren[2], Gerald Spindler[2], and Hartmut Herrmann[2]

[1] Department of Chemical, Biochemical and Environmental Engineering, University of Maryland, Baltimore County, Baltimore, 21250, USA
[2] Atmospheric Chemistry Department (ACD), Leibniz Institute for Tropospheric Research (TROPOS), Permoserstr. 15, 04318 Leipzig, Germany

[3]now at: SKW Stickstoffwerke Piesteritz GmbH, Möllensdorfer Straße 13, 06886 Lutherstadt Wittenberg

*Correspondence to*: Christopher Hennigan (hennigan@umbc.edu) or Hartmut Herrmann (herrmann@tropos.de)

**Abstract.** Aerosol acidity has importance for the chemical and physical properties of atmospheric aerosol particles and for many processes that affect their transformations and fate. Here, we characterize trends in $PM_{10}$ pH and its controlling factors over the period of 2010 – 2019 at the Melpitz research station in eastern Germany, a continental background site in central Europe. Aerosol liquid water content (ALWC) associated with inorganic species decreased by 3.4 μg m$^{-3}$ a$^{-1}$, corresponding to a 50% decrease during the analyzed time period, in response to decreasing sulfate and nitrate. Aerosol pH exhibited an increase of 0.06 units per year, a trend that was distinct from other regions. The factors controlling aerosol pH varied by season. Temperature, the most important factor driving pH variability overall, was most important in summer (responsible for 51% of pH variability) and less important during spring and fall (22% and 27%, respectively). $NH_3$, the second most important factor contributing to pH variability overall (29%), was most important during winter (38%) and far less important during summer (15%). Aerosol chemistry in Melpitz is influenced by the high buffering capacity contributed by $NH_4^+/NH_3$ and, to a lesser degree, $NO_3^-/HNO_3$. Thermodynamic analysis of the aerosol system shows that secondary inorganic aerosol formation is most frequently $HNO_3$ limited, suggesting that factors that control $NO_x$

would be more effective than $NH_3$ controls in reducing PM mass concentrations. However, the non-linear response of gas-phase $HNO_3$ and aerosol $NO_3^-$ to $NO_x$ emissions in the region, likely due to VOC controls on oxidant formation and subsequent impacts on $NO_x$ conversion to $HNO_3$, highlights the challenge associated with PM reductions needed to attain new air quality standards in this region.

## 1 Introduction

Aerosol acidity (pH) varies greatly in the atmosphere, from highly acidic sub-micron particles with pH < 0 to nascent sea spray particles with pH ~8 (Pye et al., 2020). pH is tremendously important because it affects so many atmospheric processes, including the gas/particle partitioning of weakly acidic and basic semi-volatile compounds (Ahrens et al., 2012; Nah et al., 2018; Stieger et al., 2021), aqueous-phase reaction rates, including uptake and formation of secondary organic aerosols (Tilgner et al., 2021), solubility of metals in PM (Fang et al., 2017b), optical properties of light-absorbing brown carbon (Phillips et al., 2017), nutrient deposition and bioavailability (Meskhidze et al., 2003; Baker et al., 2021), and dry deposition rates and atmospheric lifetime of reactive nitrogen species (Nenes et al., 2021). The effects of pH have direct implications for human health through their impacts on PM mass concentrations and toxicity (Song et al., 2024; Dockery et al., 1996). Aerosol pH also has direct implications for Earth's climate through the effects on PM size and number distributions, light-absorbing properties of organics, and effects on global biogeochemical cycles (Mahowald, 2011; Mahowald et al., 2018).

The pH of atmospheric particles is affected by many factors. Historically, inorganic aerosol composition – focused on the most abundant compounds – was viewed as the driving factor of pH. Sulfate, nitrate, and ammonium are certainly important contributors to aerosol pH (Pye et al., 2020); however, minor contributors to aerosol mass, such as non-volatile cations (NVCs), can also have important influences on pH (Guo et al., 2018a). Organic acids can contribute to particle acidity in clean environments (Trebs et al., 2005), but their effect on pH is likely small under more polluted conditions (Battaglia et al., 2019). The impact of relative humidity (RH) on aerosol pH has been well known because of its direct effect on aerosol liquid water content (ALWC) (Guo et al., 2015). More recently, temperature has been identified as a key factor affecting pH due to its effects on acid-base and phase equilibria (Stieger et al., 2021; Battaglia et al., 2017; Tao and Murphy, 2019; Zheng et al., 2020; Campbell et al., 2024).

Together, this indicates that a confluence of factors including regulatory issues, land use, and climate change can produce changes in regional and local aerosol pH.

Few studies have examined aerosol pH in Europe to-date. Cabauw, the Netherlands, a site near the Atlantic coast surrounded by agricultural land, had mean pH values ~3.5 across a 14-month study (Guo et al., 2018b). A remote site on the island of Crete, located in the Mediterranean Sea, typically had highly acidic aerosol pH values (~1 – 1.5), except when the biomass burning influence was high and pH was ~2.5 – 3 (Bougiatioti et al., 2016). Several studies have characterized pH at a polluted site in the Po Valley, Italy, where pH is typically ~2 – 2.5 in summer and between 3 – 4 (seasonal means) during other times of the year (Squizzato et al., 2013; Paglione et al., 2021).

Compared with the body of studies that have characterized trends in aerosol mass concentration and composition at locations globally, very few studies have characterized trends in aerosol pH through time series measurements. The only locations, to our knowledge, where trends in pH have been examined over extended time periods (> 5 years) are the southeastern USA (1998 - 2013) (Weber et al., 2016), contiguous USA (2011 - 2020) (Pan et al., 2024), continental sites in Canada (2007 - 2016) (Tao and Murphy, 2019), the Po Valley, Italy (1993 - 2018) (Paglione et al., 2021), and eastern China (2011 - 2019) (Zhou et al., 2022). Characterizing trends in aerosol pH is important because many industrialized countries have implemented regulations that have reduced emissions of $SO_2$ and $NO_x$, two key precursors to acidic inorganic aerosol components. Further, aerosol pH may also change in response to warmer temperatures and changing humidity conditions associated with climate change. Therefore, characterizing temporal trends in pH – and the factors contributing to pH – is required to better understand aerosol chemistry in these locations. Such analyses can also provide guidance for regions that are transitioning to more stringent air pollution controls – e.g., China – and regions projected to implement controls in the future – e.g., India and Pakistan. Within the present study, we characterize aerosol pH and its controlling factors over a decade at the continental background site Melpitz in central Europe. We focus on annual trends and seasonal characteristics of gas- and particle-phase composition, partitioning, meteorology, and their combined effects on aerosol pH.

## 2 Materials and Methods

### 2.1 Site description

The data presented in this analysis were collected between January 2010 – August 2019 at Melpitz, Germany (12°56'E, 51°32'N, 86 m a.s.l.), a rural background site in Saxony situated ~50 km north-east of Leipzig (Fig. S1). The research station at Melpitz is operated by the Leibniz-Institute for Tropospheric Research (TROPOS), where aerosol and gas-phase composition measurements have been conducted for more than three decades (Spindler et al., 2004; Spindler et al., 2013; Spindler et al., 2010). The site is a

part of the European Monitoring and Evaluation Programme (EMEP) network and the Aerosol, Clouds and Trace Gases Research Infrastructure (ACTRIS). The sampling site is in a rural area surrounded by flat agricultural land with no major anthropogenic sources nearby (Atabakhsh et al., 2023). The prevailing wind directions are from the west, southwest, and east (Fig. S2). The southwest wind often has a maritime origin that crosses western Europe before reaching Melpitz, while easterly winds are largely dry

continental air masses with strong anthropogenic influence from emissions in several central and Eastern European countries (Spindler et al., 2013; Stieger et al., 2018). Long-term measurements at the site show the highest $PM_{2.5}$ concentrations in air masses from east-southeast (28 µg m$^{-3}$) and lowest in those from the northwest (19 µg m$^{-3}$) (Spindler et al., 2013).

### 2.2 Gas and aerosol measurements

A Monitor for AeRosols and Gases in ambient Air (MARGA, Metrohm-Applikon, The Netherlands) system was deployed to conduct hourly measurements of water-soluble gases and inorganic ionic components in aerosols (Stieger et al., 2019; Rumsey et al., 2014; Ten Brink et al., 2007). The instrument uses a wet-rotating denuder (WRD) and a steam-jet-aerosol-collector (SJAC) to capture trace

gases (HCl, $SO_2$, $NH_3$, $HNO_3$, $HNO_2$) and water-soluble ions in particles (Cl$^-$, $NO_3^-$, $SO_4^{2-}$, $NH_4^+$, Na$^+$, K$^+$, $Mg^{2+}$, $Ca^{2+}$, and organic acid ions), respectively, at a time resolution of one hour. The sample air passes through a Teflon-coated $PM_{10}$ inlet (URG Inc., Chapel Hill, USA) and enters the WRD wetted by de-ionized water and captures water-soluble gases. The sample air then goes to a SJAC where the sample air is mixed with steam, creating a supersaturated environment that causes particles to grow rapidly into

droplets, which are subsequently collected into a liquid sample stream. The liquid solutions from the

WRD and SJAC are analyzed online using a dual anion-cation ion chromatograph. Details of the ion chromatography protocols are provided by Stieger et al. (2018). A detailed performance characterization of the MARGA system has been discussed previously (ten Brink et al., 2007; Stieger et al., 2018).

## 2.3 Aerosol pH modelling and sensitivity

Aerosol pH was calculated using the MARGA measurements, along with temperature and relative humidity (RH), as inputs to the E-AIM (Extended – Aerosol Inorganics Model) thermodynamic equilibrium model (Clegg et al., 1998; Wexler and Clegg, 2002). E-AIM was run using Model IV in batch mode, with solids formation prevented (i.e., 'metastable' mode) according to Pye et al. (2020). $H^+$ mole fraction and mole-fraction-based activity coefficients output from E-AIM were used to calculate pH using the conversion described by Pye et al. (2020). Inputs to the model were particulate inorganic ionic compounds ($Na^+$, $NH_4^+$, $Cl^-$, $NO_3^-$, and $SO_4^{2-}$), temperature, and RH. The $Cl^-$ and $NO_3^-$ inputs represented the total gas + aerosol concentrations ($HCl + Cl^-$ and $HNO_3 + NO_3^-$, respectively), though aerosol $NH_4^+$ and $NH_3(g)$ were separate inputs to the model. E-AIM does not include the non-volatile cations (NVCs) $K^+$, $Mg^{2+}$, and $Ca^{2+}$. Therefore, $K^+$, $Mg^{2+}$, and $Ca^{2+}$ were input as $Na^+$ equivalents. The measurement values of NVCs that were below the method detection limit (MDL) were assigned the value of the MDL for model runs. For $Na^+$, $Ca^{2+}$, $Mg^{2+}$ and $K^+$, those values were 0.02, 0.01, 0.01 and 0.01 µg m$^{-3}$, respectively. Guo et al. (2018a) showed that this assumption affects our pH calculations by < 0.1 pH unit (i.e., compared to simulations using zero for NVC inputs when measurements were below the MDL), in agreement with other studies that have shown a minor effect of NVCs on pH when present in low concentrations (Battaglia Jr et al., 2021; Tao and Murphy, 2021; Zheng et al., 2020). $H^+$ and $OH^-$ ions are required inputs to the model and were estimated using the total ion charge balance fulfilling the requirement of electroneutrality. Thermodynamic calculations of pH are challenged at low RH (Pye et al., 2020). Similarly, the operating temperature range for E-AIM model IV with $NH_4$ and $Cl^-$ or $Na^+$ and other ions is 263.15 – 330 K (https://www.aim.env.uea.ac.uk/aim/aim.php). Therefore, samples with both T < 264 K and RH < 60% were excluded from the present analysis (1% and 16% of observations, respectively). Further, data with modeled pH outside the range of -0.5 – 6.0 and ALWC outside the range of 0.34 – 500 µg m$^{-3}$ were outliers and were also excluded (3% of observations). Finally, the model was

not run if there were any data missing from the aerosol, gas-phase, or meteorology measurements (e.g., for instrument maintenance, calibrations, QA/QC checks, 20% of observations). Overall, with these conditions applied, we modeled 54614 out of a possible 86185 hourly data points (~63%) across the 10-year study, with $n = 13130$ in spring (March, April, and May), $n = 11350$ in summer (June, July, and August), $n = 14200$ in autumn (September, October, and November), and $n = 15934$ in winter (December, January, and February).

The E-AIM model was run without organics as inputs, a good assumption that has minimal effect on the predicted pH under the imposed RH limitation (Battaglia et al., 2019). Organic aerosol components can take up water and contribute substantially to ALWC, so the results presented herein do not account for the fraction of ALWC associated with those organic species. Based on AMS measurements of $PM_1$ composition at the Melpitz research station, it is likely that water content associated with organics was ~20% of the total ALWC assuming an average kappa value of 0.15 for the organics (Atabakhsh et al., 2025). Although the model output represents ALWC associated with $PM_{10}$, Kakavas et al. (2021) showed that ALWC was many times higher in fine particles than coarse mode particles across Europe. The approximations and assumptions are evaluated through the measured and predicted $NH_3$ partitioning, a key metric used to assess thermodynamic model calculations of aerosol pH. $NH_3$ (slope = 1.006, intercept = 0.15, $R^2$ = 0.99, $n = 54617$) and $NH_4^+$ (slope = 0.933, intercept = 0.22, $R^2$ = 0.928, $n = 54617$) showed excellent agreement between the predictions and measurements (Fig. S3), indicating the model assumptions employed were valid for this dataset.

The approach of Tao and Murphy (2021) was used to investigate the contribution of different factors to pH variability in Melpitz. The method uses pH calculated according to $NH_3$ partitioning theory (Eq. 1) to identify the factors responsible for the difference in pH between two systems, whether that be different locations or different times at the same location. The framework of Tao and Murphy (2021) assesses the following factors for their influence on $NH_3/NH_4^+$ phase partitioning, and hence pH: (1) temperature, (2) gas-phase $NH_3$ concentration, (3) RH, (4) particle properties, and (5) an error term associated with simplifying assumptions in the derivation. Temperature affects equilibrium constants ($K_H$, $K_a$, and $K_w$) that affect $NH_3/NH_4^+$ phase partitioning. Duan et al. (2025) demonstrated that T effects on activity coefficients and semi-volatile vapor pressures also contribute to its overarching influence on pH.

The $NH_3$ (g) abundance clearly affects $NH_3$ partitioning, as a $NH_3/NH_4^+$ system at equilibrium will shift towards the gas phase in response to a reduction in $NH_3$ and towards the particle phase in response to an $NH_3$ increase. RH directly affects ALWC, which impacts the particle-phase activities of aerosol species, including $NH_4^+$ and $H^+$, both of which directly factor into the computation of $NH_3/NH_4^+$ phase partitioning. Finally, particle properties account for the particle hygroscopicity, which also influences ALWC, and the ammonium activity coefficient, which affects the ammonium activity.

In the present analysis, we used the average study conditions in Melpitz to calculate the factors responsible for seasonal variability in pH. The approach decomposes the pH difference between two points into the difference between key factors that control or influence pH, Equations 2 and 3 (taken from Eq. 8 and Eq. 4 in Tao and Murphy (2021)):

$$pH = log_{10}\left[\frac{n(NH_3) \cdot K_H \cdot T \cdot R}{\chi(NH_4^+) \cdot \gamma(NH_4^+) \cdot 55.509 \cdot P_{atm}}\right] \tag{1}$$

$$pH_2 - pH_1 = \Delta log_{10}[n(NH_3)] + \Delta log_{10}\left[\frac{\kappa \cdot M(NH_4^+)_{avg}}{\rho(NH_4^+)_{avg} \gamma(NH_4^+)}\right] + \tag{2}$$

$$\Delta log_{10}\left[\frac{RH}{1-RH}\right] + \Delta log_{10}[K_H \cdot T] + \Delta(\varepsilon)$$

$$pH_2 - pH_1 = \Delta f(NH_3) + \Delta f(particle\ properties) + \Delta f(RH) + \Delta f(T) + \Delta(\varepsilon) \tag{3}$$

where $n(NH_3)$ is gas-phase molar concentration of ammonia (mol m$^{-3}$), $\kappa$ is the particle hygroscopicity parameter, $M(NH_4^+)_{avg}$ is equivalent molecular weight of ammonium in the aerosol phase (g mol$^{-1}$), $\rho(NH_4^+)_{avg}$ is the density of particulate ammonium salt (kg m$^{-3}$), $\chi(NH_4^+)$ is the mole fraction of ammonium, $\gamma(NH_4^+)$ is the mole fraction-based activity coefficient of ammonium, $K_H$ is Henry's constant for $NH_4^+$ salt (See equation 4, Tao and Murphy (2021)), $RH$ is the relative humidity, $T$ is the temperature (K), $PP$ represents particle properties (a parameter that combines the influence of particle composition and concentration), and an error term ($\Delta\varepsilon$) that accounts for simplifying assumptions used in

the calculation of pH (the term increases under low RH conditions (RH <50%)). The *PP* term includes the measured ratios of ammonium to sulfate and nitrate, as well as the E-AIM modeled $\gamma(NH_4^+)$ and $\kappa$. The values for the parameters in equation (1) were obtained from the results of the E-AIM thermodynamic model. In this case, $pH_1$ is the average pH for the entire 10-year (2010 – 2019) study in Melpitz, while

$pH_2$ is the modelled aerosol pH at a given time. The difference in pH between a single observation and the overall study average can then be attributed to the different factors given in Eq. 3. The error term was low ($\Delta\varepsilon$ ~1-3% overall and within each season), suggesting the simplifying assumptions used in the definition of pH for this analysis were appropriate (Tao and Murphy, 2021). The pH calculated by $NH_3$ partitioning showed excellent agreement with the E-AIM pH (slope = 1.02, $R^2$ = 0.994, Fig. S4),

suggesting that the method provides useful insight into the factors that affect pH in Melpitz.

We also applied the multiphase buffer theory developed by Zheng et al. (2020) to quantify the buffering capacity in Melpitz attributed to different compounds. The buffering capacity, β, represents the ratio between the mass of acid (base) added to a system and the resulting decrease (increase) in aerosol pH. Higher values of β indicate systems that will have relatively smaller changes in pH for a given amount

of acid or base added. Multiple compounds present in the atmosphere contribute to the aerosol buffering capacity, including the pairs $NH_3$-$NH_4^+$, $HNO_3$-$NO_3^-$, and $HSO_4^-$-$SO_4^{2-}$ as well as NVCs and $H_2O$. The concentrations of each buffering species or pair, as well as meteorological conditions, determine the overall β as well as the contribution of each acid-base pair to the buffering capacity. The calculation, based on Zheng et al. (2020) is:

$$\beta = \frac{dn_{base}}{dpH} = 2.303 \left\{ \frac{K_w}{[H^+]} + [H^+] + \sum_i \frac{K_{a,i}^*[H^+]}{(K_{a,i}^* + [H^+])^2} [X_i]_{tot}^* \right\} \qquad (4)$$

Where $n_{base}$ is the molal concentration (mol kg$^{-1}$), pH is the aerosol pH, $K_w$ is the temperature-dependent water dissociation constant, $K_{a,i}^*$ is the effective acid dissociation constant for species i, $[H^+]$ is the molality based aqueous concentration of H$^+$ (mol kg$^{-1}$), and $[X_i]_{tot}^*$ is the total equivalent molality-based concentration of conjugate acid-base pair i (e.g., $[X_i]_{tot}^*$ for $NH_4^+$/$NH_3$ would include [NH$_4^+$(aq)],

[NH$_3$(aq)], and [NH$_3$(g)], with all concentrations expressed as moles per kg water).

# 3 Results

## 3.1 Annual trends of aerosol pH

There were notable trends in several major species measured at Melpitz (Fig. 1, Table 1). The trend in annual sulfate concentrations was -0.15 $\mu g\ m^{-3}\ a^{-1}$, a statistically significant decrease that corresponded to a ~60% reduction from 2010 to 2019 (Fig. 1a). The trend in annual Tot-NO$_3$ concentrations was also -0.15 $\mu g\ m^{-3}\ a^{-1}$, a statistically significant decrease that corresponded to a ~50% reduction from 2010 to 2019 (Fig. 1b). These trends are the result of ongoing reductions in precursor SO$_2$ and NO$_x$ emissions across continental Europe (Turnock et al., 2015; Vestreng et al., 2009; Hamed et al., 2010; Jonson et al., 2022). The observed trends in aerosol sulfate are consistent with the broader trends of decreasing sulfate across central Europe that have occurred since the 1980s (Vestreng et al., 2007). In response to the decreases in SO$_4^{2-}$ and Tot-NO$_3$, aerosol NH$_4^+$ concentrations exhibited a similar decrease of ~60% from 2010 to 2019. Sulfate, nitrate, and ammonium (SNA) are the most abundant inorganic aerosol components in Melpitz (Spindler et al., 2013), and thus, they control ALWC abundance. Concentrations of ALWC decreased by 3.4 $\mu g\ m^{-3}\ a^{-1}$ (Table 1), representing a ~50% decrease from 2010 to 2019 (Fig. 1c) because of the 50 – 60% decreases in SNA.

Although the concentration of NH$_4^+$ decreased, the average Tot-NH$_3$ concentration remained relatively flat from 2010 – 2019 (Fig. 1d), indicating a shift in NH$_3$ partitioning towards the gas phase. The annual trend in the Tot-NH$_3$ concentration was not statistically significant at the 95% confidence level (Table 1). A moderate increase in total column NH$_3$ was observed over Germany for the period roughly corresponding to the present study, 2008 – 2018, although most of the observed increase occurred in 2017 and 2018 (Van Damme et al., 2021). NVCs showed a slight positive trend from 2010 – 2019 (Fig. 1e), +0.01 $\mu g\ m^{-3}\ a^{-1}$, but the increase was not statistically significant at the 95% confidence level.

The annual average aerosol pH varied between 2.76 – 3.45 and followed a gradual linear increase of ~0.06 pH units per year, leading to a 0.6 pH unit rise over the past ten years (Fig. 1f). The increasing trend in pH is statistically significant at the 95% confidence level (Table 1), demonstrating a decrease in particle acidity in Melpitz. The increasing pH trend in Melpitz is interesting to contrast with trends in other locations because it is the only site where particle acidity has shown a significant decrease (pH increase). Weber et al. (2016) found a slight decrease in pH in the southeastern United States over 1998

– 2013, though the trend appears flat from 2006 – 2013. Tao and Murphy (2019) observed no significant

changes in pH in five Canadian cities from 2007 – 2016, though acidity in Ottawa increased (pH decreased). Paglione et al. (2021) used fog water composition measurements to estimate annual trends in aerosol pH in the Po Valley, Italy. While fog water pH has steadily increased since 1993, rising by 0.04 units per year on average as a result of decreasing emissions of $SO_2$ and $NO_x$, calculated aerosol pH declined by ~1.5 units from 1993 – 2018 (Paglione et al., 2021). Although the trends in aerosol pH in

Melpitz appear contrary to those in North America and the Po Valley, the results are highly consistent with trends in fog water and cloud water pH observed over central Europe (Pye et al., 2020). Using composite results from dozens of sites across Europe, cloud and fog water pH has increased by 0.56 pH units per decade since 1980 (~2.2 pH unit increase from 1980 to 2020) (Pye et al., 2020). This closely matches the 0.6 pH unit increase in Melpitz aerosol for 2010 - 2019. Further, pH measured in cloud water

at Whiteface Mountain, NY, has shown a marked increase over a similar timescale, suggesting key differences in the thermodynamic regime of the aerosol in North America (Lawrence et al., 2023).

## 3.2 Seasonal trends in aerosol pH, ALWC, and composition

Aerosol composition in Melpitz is characterized by large gradients in the seasonal concentrations

of major components. Median $SO_4^{2-}$ concentrations were relatively consistent between seasons, with a range of only 1.14 – 1.37 µg m$^{-3}$. However, the quartile and 90$^{th}$ percentile values indicate relatively frequent enhancements in $SO_4^{2-}$ concentrations during winter and similarly infrequent enhancements during summer (Fig. 2a). Under the coldest conditions, Melpitz was under heavy influence from emissions to the east and southeast (Poland, Czech Republic, and Austria) (Stieger et al., 2018). Sulfate showed an

inverse relationship with temperature during winter (Fig. S5) due to higher energy demand and increased coal combustion in eastern Europe for domestic heating (Atabakhsh et al., 2023; Hamed et al., 2010). Tot-$NO_3$ concentrations exhibited a similar seasonal pattern as $SO_4^{2-}$, with the highest median concentrations in winter (3.41 µg m$^{-3}$), followed by spring (2.95 µg m$^{-3}$), fall (2.53 µg m$^{-3}$), and then summer (1.84 µg m$^{-3}$, Fig. 2b). Average Tot-$NO_3$ concentrations in winter and spring were approximately

double the summer average, while differences in the upper quartile and 90$^{th}$ percentile values were even larger. The seasonal Tot-$NO_3$ pattern is also apparent in Fig. 2b, where maxima in winter and minima in

summer occurred almost annually. The seasonal Tot-NO$_3$ results reflect higher energy demand and increased coal combustion under colder conditions in the region, as well as differences in the air mass source regions and boundary layer structure.

In contrast to SO$_4^{2-}$ and Tot-NO$_3$, mean and median concentrations of Tot-NH$_3$ were ~1.5 times higher in the summer and fall compared to winter, and 2.3 times higher in spring (Fig. 2c). Tot-NH$_3$ concentrations in the spring regularly exceeded 10-15 µg m$^{-3}$, an infrequent occurrence in other seasons (Fig 2c). Melpitz is located in eastern Saxony, a region dense with farmland. The elevated Tot-NH$_3$ observed in spring is attributed to fertilizer applications at the beginning of the growing season, as this is a major source of NH$_3$ emissions in the region (Viatte et al., 2022). The observations in Melpitz are consistent with those in other parts of Europe, where NH$_3$ emissions peak in March, April, and May due to fertilizer spreading (Van Der Graaf et al., 2022). The region surrounding Melpitz shows large model-measurement differences in NH$_3$ emissions, and predictions are worse during the spring (van der Graaf et al., 2022).

Tot-NH$_3$ concentrations showed a clear relationship with temperature in all seasons except winter (Fig. 3a). Due to the strong effect of temperature on NH$_3$ partitioning (Fig. 3d), mean NH$_3$(g) concentrations increased with temperature in each season (Fig. 3b). Mean gas-phase fraction $\varepsilon_{NH3}$ ($\varepsilon_{NH3} =$ NH$_3$(g)/(NH$_3$(g) + NH$_4^+$)) values showed highly consistent behavior with temperature across seasons despite changing aerosol composition. The results in Fig. 3 suggest that agricultural emissions represent the major source of Tot-NH$_3$ during most seasons in Melpitz, consistent with broader studies across Europe (Sutton et al., 2013; Backes et al., 2016a). Very strong correlations ($r^2$ ~0.8) between NH$_3$(g) and skin temperature (physical temperature of the earth's surface) – both remotely sensed – are observed in this area (Viatte et al., 2022). We observed similar temperature-NH$_3$ relationships within each season, but the annual results showed less consistency. We observed stronger seasonal gradients, especially much higher spring concentrations, than the adjacent area (lon [11, 12], lat [50, 52]). However, it should be noted that the IASI satellite results are based on twice-daily total column measurements, at 09:30 and 21:30 local time, and it is not straightforward to convert between total column NH$_3$ and local, in situ concentrations: this relationship varies by region (van der Graaf, ACP, 2022; Viatte et al., 2022).

The seasonal profile of ALWC was distinct from the other species plotted in Fig. 2. Mean values of ALWC were a factor of 3-4 higher than the median each season, reflecting skewed distributions. Median RH values were 74.7% (summer), 79.0% (spring), 89.2% (fall), and 89.6% (winter). ALWC increases exponentially with RH, so RH values above 90%, and especially above 95%, cause this observation. Approximately 28% of observations in the winter and 32% of observations in fall had an RH > 95% (Fig. S6), reflecting the humid conditions experienced in Melpitz during this time of year. Conversely, approximately 17% of observations in summer and 16% of observations in spring had an RH > 95% (Fig. S6). This explains why the mean, upper quartile, and 90[th] percentile ALWC concentrations were so much higher in winter and fall than in spring and summer. Similar skewed seasonal distributions in ALWC, with mean seasonal values near or exceeding upper quartile values, were also observed in a decadal study in Shanghai, China (Zhou et al., 2022).

Aerosol pH showed seasonal trends that reflect the combined influence of inorganic composition and meteorology (Fig. 2d). Aerosol pH was highest in spring with mean and median pH of 3.55, coinciding with the maxima in Tot-$NH_3$. Lowest mean and medial pH values were observed during summer (2.83 and 2.89, respectively), when highest temperatures and lowest ALWC levels were observed. It is likely that our imposed RH limit (> 60%) for pH calculation removed a subset of the summer data with very low pH values (< 2) because of the strong effect of ALWC on pH. Therefore, it is more likely that the summer data exhibit lower quartile and 10[th] percentile values than Fig. 2d shows. Winter had mean and median pH values (3.04 and 3.21, respectively) that reflect the high sulfate and low Tot-$NH_3$ concentrations. Aerosol pH in fall exhibited the second highest season mean and median values (3.30 and 3.33, respectively). The pH in Melpitz showed a very different seasonal pattern from that observed in six Canadian cities, where pH was a minimum in summer and maximum in winter, with spring and fall in between as pH transitioned between the extremes (Tao and Murphy, 2019). As discussed above in Section 3.1, the thermodynamic drivers of pH Canada and the associated trends in pH appear to be different from those in Melpitz. Detailed analysis and discussion of the factors contributing to pH variability in Melpitz are presented in Section 3.4.

## 3.3 Diurnal trends in aerosol pH

Figure 4 shows the mean diurnal profiles of key species and parameters separated by season. Aerosol pH exhibited a maximum in the early morning hours (04:00-06:00, Fig. 4g) that coincided with the peak ALWC (Fig. 4a). As temperature increased throughout the morning and into the afternoon (Fig. 4b), ALWC and pH decreased. Daily minima in ALWC and pH occurred in the afternoon, coincident with the peak temperature, but rose in the afternoon and evening as temperature decreased. The diurnal profiles of ALWC and pH are qualitatively consistent with those from diverse locations, including the southeastern U.S. (Guo et al., 2015), Baltimore, MD (Battaglia et al., 2017), California (Guo et al., 2017), and Beijing (Ding et al., 2019). This reflects the strong response of pH to meteorological inputs, namely temperature and RH (Battaglia et al., 2017; Zheng et al., 2020; Tao and Murphy, ES&T, 2021). The pH profiles in Fig. 6 are consistent with those Fig. 3d, which shows mean pH was lowest in summer, followed by winter, fall, and spring. Likewise, the ALWC profiles in Fig. 4 are consistent with mean values shown in Fig. 2e, with summer having the lowest ALWC followed by spring, winter, and fall.

The diurnal profiles of Tot-$NH_3$, Tot-$NO_3$ , and sulfate show different processes and sources contributing to their seasonal concentrations (Fig. 4d, e, and f, respectively). Sulfate peaked between 12:00 – 15:00 daily (local), with minima at night, reflecting secondary formation processes. The midday peak during winter is weaker than during other seasons, reflecting the increased importance of regional transport during this time (Stieger et al., 2018). Tot-$NO_3$ increased in the morning and peaked between 07:00 – 10:00 daily due to photochemical $HNO_3$ formation. The diurnal profiles of Tot-$NO_3$ were qualitatively similar during spring, summer, and fall, though the concentrations were quite different, consistent with Fig. 2b. Tot-$NH_3$ exhibited very different behavior than sulfate or Tot-$NO_3$. During spring, summer, and fall, Tot-$NH_3$ concentrations strongly increased in the morning with the onset of sunrise and the daily temperature increase, approximately doubling between 04:00 and 09:00 each day. Dew is an important nighttime sink and morning reservoir for HONO in Melpitz (Ren et al., 2020). Therefore, in addition to fresh emissions, dew evaporation may be an important source of morning $NH_3$ (Wentworth et al., 2016). RH peaks at night and in the early morning and the frequency of high RH periods suggests dew regularly forms around Melpitz (Fig S6). While Tot-$NH_3$ emissions vary considerably by season, Fig. 3d shows that $NH_3$ partitioning is consistent across seasons and is mostly regulated by temperature.

ALWC shows similar diurnal profiles between seasons, with maxima around 04:00 – 06:00 and minima around 14:00 – 16:00 (Fig. 4a). The seasonal ALWC profiles are qualitatively consistent with those observed in other locations, including North America, China, and other locations in Europe (Pye et al., 2020). Aerosol pH profiles are also qualitative similar each season, with maxima at night and early morning and minima in the afternoon, due to the concentrating effects of the reduced ALWC (Fig. 4g). Although the seasonal profiles also follow the general behavior observed in many other locations, the amplitude was somewhat less, as the range in diurnal profiles was only ~0.5 pH units in summer and spring and only ~0.3 units in winter and fall. This suggests a higher aerosol pH buffering capacity. The calculated aerosol pH values are consistent with a modeling study of size-resolved aerosol pH over Europe (Kakavas et al., 2021). In May, aerosol pH predictions around Melpitz were ~3 for $PM_1$ and ~4 for $PM_{1-2.5}$, consistent with our observations. The simulations capture well the diurnal trends; however, they overestimate the range in daily pH, predicting ~2-2.5 unit change in pH throughout the day (Kakavas et al., 2021).

Fig. 5 shows the aerosol buffering capacity calculated according to Zheng et al. (2020). Overall, aerosols in Melpitz have a high buffering capacity compared to other locations like the southeast USA (Zheng et al., 2020). $NH_4^+/NH_3$ contributes most of the buffering capacity in Melpitz, followed by $NO_3^-/HNO_3$. The relative order of importance of buffering pairs was generally consistent across seasons, as well (Fig. S7). Other species were relatively minor contributors to the buffering capacity because of the observed aerosol pH range. The high buffering capacity leads to the relatively modest amplitude in diurnal pH values observed in Fig. 6a. By contrast, the eastern USA has a much smaller buffering capacity than central Europe, contributing to diurnal averages of aerosol pH that vary by 1 – 2 pH units throughout the day (Nah et al., 2018; Guo et al., 2015). Pye et al. (2020) found that diurnal profiles of aerosol pH in Melpitz show lower amplitude than similar profiles in a comparison to diverse environments. Overall, the high buffering capacity of aerosols in Melpitz suggests that pH may undergo more modest changes in response to climate change because of the high Tot-$NH_3$ and its contribution to buffering capacity. Further, analysis of pH across diverse regions of the globe revealed that $NH_3$-$NH_4^+$ is the dominant buffering pair in the atmosphere, highlighting the important role of agricultural emissions in affecting aerosol chemistry (Zheng et al., 2020).

It should be noted that the present study characterizes the pH of $PM_{10}$ but small particles are systematically more acidic than large particles. Kakavas et al. (2021) modeled aerosol pH over continental Europe during summer in four size bins: $PM_1$, $PM_{1-2.5}$, $PM_{2.5-5}$, and $PM_{5-10}$. They found that smaller size bins are systematically more acidic than larger size bins due to the higher presence of non-volatile cations from sea salt and dust in larger particles, consistent with prior studies (Keene et al., 2004; Fridlind and Jacobson, 2000; Fang et al., 2017a). The average pH of $PM_1$ was 1.3 pH units lower than the largest coarse bin ($PM_{5-10}$), however, this difference exhibited a strong spatial dependence. Coastal areas with high sea salt influence, such as Denmark and northwestern France, showed the largest pH differences between different size fractions. Kakavas et al. (2021) specifically analyzed conditions in Melpitz and found the pH differences between size fractions to vary diurnally. Throughout the day, $PM_{5-10}$ was ~1 pH unit higher than $PM_1$; however, the differences between $PM_1$ and $PM_{1-2.5}$ were as large as ~1 pH unit at night but as small as ~0.25 pH unit in the daytime. Very similar differences and diurnal patterns were found for the $PM_1$ vs. $PM_{2.5-5}$ fractions. We expect the simulations of Kakavas et al. (2021) to represent an upper bound on pH differences between fine and coarse fractions in Melpitz because their simulations were specific to summer: Spindler et al. (2013) showed through analysis of long-term aerosol composition data that dust concentrations in Melpitz are highest during summer. Therefore, we expect the present results to inform the trends of aerosol pH in smaller size fractions ($PM_1$ and $PM_{2.5}$), with pH values ~0.25-1 unit lower, depending on the season and time of day.

**3.4 Drivers of pH variability**

The factors contributing to pH variability in Melpitz are shown in Fig. 6. Overall, temperature and RH together accounted for 45% of pH variability, relative to the study average. This is qualitatively

consistent with prior studies that have identified the important role of meteorology in controlling pH (Battaglia et al., 2017; Zheng et al., 2020; Tao and Murphy, 2021). $NH_3$ was the second most important factor, responsible for 29% of pH variability over the entire study, followed by particle properties, which contributed 23% of the variability in pH. The role of $NH_3$ in Melpitz was similar to its influence on pH variation in Toronto (25%, Tao and Murphy, 2021). Note that this analysis does not capture the directional change in pH resulting from each factor. In fact, a given factor can cause both positive and negative deviations from the average, depending on conditions. Because the method quantifies changes relative to the mean study conditions, lower temperatures observed in winter contribute to increases in pH while higher temperatures in summer contribute to decreases in pH. This can be seen in Equations 1 and 3, as well as the effects of temperature on $NH_3$ partitioning (Fig. 3d). In this analysis, these opposing effects do not cancel out in quantifying the factor's contribution to pH variability.

Consistent with the results in Fig. 2, the factors contributing to pH variability also showed distinct seasonal differences (Fig. 6). Temperature was the most significant contributor to pH variability overall, but its influence ranged from 51% in summer to only 22% in spring. Likewise, $NH_3$ – the second most important factor overall – ranged from 38% in winter to 15% in summer. It is important to note that the influence of $NH_3$ on pH variability does not correlate with the Tot-$NH_3$ concentration. For example, summer and fall Tot-$NH_3$ concentrations were quite similar (Fig. 2c and Fig. 4d), yet $NH_3$ had more influence on pH variability during fall than summer (22% vs. 15%, respectively). Particle properties, while the third most important factor overall, had the most important effect on pH variability in fall (32%). RH contributed 10% – 16% of pH variability in all seasons, far less than temperature during winter and summer, but closer to the effect on temperature during spring and fall.

It should be noted that that the various factors contributing to pH variability are not completely independent. For example, temperature and $NH_3$ often influence pH in opposite directions, but $NH_3$ emissions and abundance have a clear dependence on temperature (Fig. 3). Similarly, temperature and RH are typically inversely related at ground level. RH affects ALWC, which changes solute concentrations in aqueous particles and gas-particle partitioning of semi-volatile compounds (Ansari and Pandis, 1999). Temperature is a master variable that affects most processes in the environment – with regards to aerosol pH, it affects compound vapor pressures, equilibrium constants, activity coefficients,

solubilities (i.e., overall partitioning), and reaction rates (Duan et al., 2025; Pye et al., 2020; Tilgner et al., 2021). Therefore, while their effects on aerosol pH can be separately quantified, these factors can somewhat offset each other in the ambient atmosphere relative to conditions where one factor changes independently of the others. A similar phenomenon was observed and noted for the Canadian data set
analyzed by Tao and Murphy (2021).

The results in Fig. 6 also highlight interesting trends that provide additional context to the seasonal and diurnal data (Figs. 2 and 4). For example, while the pH values in winter and summer were similar (mean values differ by only 0.2 units), the factors contributing to pH variability were distinctly different. During winter, $NH_3$ was a more important driver of pH variability than temperature (38% and 32%,
respectively), but temperature was far more important during summer (51% to 15%). Conversely, pH values in spring and fall were also similar (mean values differ by 0.2 units), but the drivers of pH variability were more similar than summer-winter.

The present results are consistent with a study investigating the drivers of pH variability in Canada, where temperature was the most significant factor, followed by $NH_3$ (Tao and Murphy, 2019;
2021). The observations in Canada also found temperature to be a much more important factor in summer and winter than it was in spring and fall, consistent with the results in Melpitz. However, particle properties had a much greater influence on pH variability in Melpitz, ranging from 19% in winter to 32% in fall, than Canada, where it contributed < 10% to pH variability in each season. Tao and Murphy (2021) showed that $\gamma(NH_4^+)$ and $\kappa$ were the two most important individual particle property factors contributing
to pH variability. The results in Melpitz and Canada also differ significantly from a study that characterized pH variability at three coastal sites in China, where temperature was only a minor contributor (8 – 12%) to pH variability (Wang et al., 2022). There, ammonia was a major factor in driving pH variability (23 – 42%), similar to Melpitz and Canada, but RH exhibited a much stronger influence (25 – 39%) than it did in Melpitz and Canada. There are several likely reasons for the observed differences
between Melpitz and Canada and the Chinese sites. The Chinese measurements only spanned 3 – 5 weeks at each site whereas the studies in Melpitz and Canada each included ~10 years of data. It is likely that similarly brief periods of time in either location would also yield factor contributions that significantly differ from seasonal and annual averages. The coastal Chinese sites also experienced less daily

temperature variation – only ~2 – 3 °C in daily temperature range – than either the Canadian or German studies, likely contributing to the much smaller effect of temperature on pH variability (Wang et al., 2022).

## 3.5 Sensitivity of PM mass to precursor availability

Aerosol concentrations always respond to emissions of non-volatile components and precursors, such as NVCs and sulfate. For semi-volatile components, especially $NH_3$ and $HNO_3$, thermodynamic conditions dictate whether aerosol mass responds to changing concentrations. Following the approach of Nenes et al. (2020), we calculated the sensitivity of $PM_{10}$ mass concentrations to $HNO_3$ and $NH_3$ availability. The approach defines regimes where the aerosol mass is limited by $HNO_3$ and $NH_3$ availability, mapped in pH and ALWC space. Aerosol mass can be sensitive to $NH_3$, $HNO_3$, both species, or neither: temperature, ALWC, and pH are the key parameters that determine the thermodynamic regime for a given set of conditions (Nenes et al., 2020). Fig. 7 shows the different thermodynamic regimes by season calculated using the mean temperature within each season (275 K in winter, 281 K in spring, 291 K in summer, and 282 K in fall). $NH_4NO_3$ concentrations in Melpitz are typically high, accounting for ~30% of annual $PM_{2.5}$ and $PM_{10}$ mass in the region (Spindler et al., 2013). Therefore, if a sensitivity to $NH_3$ or $HNO_3$ (or both) is identified, this suggests the aerosol concentration would respond to changes in these semi-volatile precursors. In the present analysis, we only considered ALWC associated with inorganic aerosol components. While water-soluble OA likely exerts minor effects on pH (Battaglia et al., 2019), it can contribute to the ALWC. Consideration of the ALWC contributed by inorganic and organic compounds would result in a minor shift of the individual data points in Fig. 7, although the logarithmic scale indicates very few points would shift to a different thermodynamic regime (e.g., from insensitive to $HNO_3$-sensitive).

Aerosol concentrations in Melpitz were most often sensitive to $HNO_3$, ranging from 47% in winter to 83% in spring (Figs. 7 and 8). Very few observations showed sensitivity to $NH_3$, alone, with only 0.2% at a minimum (spring) and 2.6% maximum (summer). Many observations showed sensitivity to both $NH_3$ and $HNO_3$, from 16% in spring to 51% in winter. Therefore, the fraction of observations sensitive to $HNO_3$ (either exclusively or with $NH_3$) was 94% in summer and > 99% in spring and fall. Less than 1%

of observations showed insensitivity to $NH_3$ and $HNO_3$ in winter, spring, and fall, and only 3.3% of observations in summer showed insensitivity to both species (Figs. 7 and 8). Temperature exerts a controlling effect on the partitioning coefficients of $NH_3$ and $HNO_3$ (affecting the intercept of the red and blue lines in Fig. 7), with higher temperatures increasing the region insensitive to both $NH_3$ and $HNO_3$ (white region in Fig. 7). This explains why the summer had more than an order of magnitude more observations in the insensitive regime, although this was still a very minor fraction of the overall observations in this season. For a given combination of temperature and pH, higher ALWC increases the condensed-phase fraction of $NH_3$ and $HNO_3$, leading to conditions where the aerosol is sensitive to one or both of $NH_3$ and $HNO_3$. Lower ALWC leads to the opposite effect, decreasing the sensitivity of aerosol mass to either $NH_3$ or $HNO_3$. Thus, the generally high RH conditions in Melpitz (Fig. S6) and high ALWC levels push a lot of the observations into thermodynamic regimes sensitive to one or both of $NH_3$ and $HNO_3$. This analysis offers insight into changes that may occur under a changing climate, as well. Higher temperatures will result in more acidic particles and lower ALWC (Battaglia et al., 2017). As shown in Fig. 7, this will make the aerosol mass less sensitive to changes in $NH_3$ and $HNO_3$, offering fewer mitigation options for controlling PM levels.

The results in Figs. 7 and 8 suggest strategies to reduce $PM_{10}$ concentrations in the region should prioritize $NO_x$ emissions reductions over $NH_3$ controls because it is the key $HNO_3$ precursor. This result is consistent with a one-year study in Cabauw, Netherlands, which also found $NO_x$ controls likely to reduce aerosol $NO_3^-$ more than $NH_3$ controls (Guo et al., 2018b; Nenes et al., 2020). Pay et al. (2012) conducted a modeling study over Europe and found results consistent with the present analysis – more frequent aerosol regimes limited by $HNO_3$ availability, suggesting $NO_x$ controls may be more effective at reducing $PM_{2.5}$. Another study modeled inorganic aerosol formation over Germany and found the aerosols insensitive to $NH_3$ reductions, especially during the spring when $NH_3$ emissions peak (Renner and Wolke, 2010). Both studies are consistent with the thermodynamic framework shown in Figs. 7 and 8. Backes et al. (2016b) modeled the response of aerosol concentrations over western Europe to changes in $NH_3$ emissions and found that decreases in $NH_3$ emissions led to decreases in $PM_{2.5}$ and $PM_{10}$ mass due to decreases in $NH_4NO_3$. In strong agreement with our thermodynamic analysis, aerosol mass showed the largest sensitivities to $NH_3$ reductions in winter (Backes et al., 2016b).

Aerosol $NO_3^-$ formation from $NO_x$ is highly non-linear, suggesting limitations to the effectiveness of $NO_x$ controls on secondary aerosol formation. For example, if a location falls under the $HNO_3$ limited thermodynamic regime (or both $HNO_3$ and $NH_3$ limited), reductions in $NO_x$ do not necessarily translate into PM reductions the way that $SO_2$ reductions lead directly and almost linearly to reductions in aerosol sulfate. In the eastern US, there was a very weak response of aerosol nitrate to $NO_x$ reductions due to simultaneous changes in $SO_2$ emissions and changes in aerosol pH shifted the $NO_3^-/HNO_3$ partitioning towards the particle phase (Shah et al., 2018). So, although $NO_x$ reductions may decrease Tot-$NO_3$, even minor changes in $NO_3^-/HNO_3$ partitioning can keep $NO_3^-$ relatively constant, or even lead to an increase (Ansari and Pandis, 1998). A similar phenomenon was also observed in the Salt Lake Valley, Utah, where wintertime ammonium nitrate formation did not respond to $NO_x$ controls, but was instead sensitive to VOC emissions (Womack et al., 2019). This occurs because $NO_x$ conversion to $HNO_3$ is oxidant-limited in some environments. In such cases, control of VOC emissions, rather than $NO_x$ or $NH_3$, can provide the most direct effect on aerosol $NH_4NO_3$ (Dang et al., 2023; Womack et al., 2019). Changes in $NO_3^-/HNO_3$ partitioning can also reduce the $HNO_3$ dry deposition sink, increasing the atmospheric lifetime of Tot-$NO_3$ (Zhai et al., 2021). This complex chemistry is consistent with a recent study that found that a 23% reduction in $NO_x$ emissions during the COVID-19 lockdown period was associated with only a ~5% decrease in $PM_{2.5}$ mass in Germany (Balamurugan et al., 2022). Further, actual emissions control measures need to account for costs and analysis by Liu et al. (2023) indicates $NH_3$ controls are 5-10 times more cost effective than $NO_x$ controls at reducing PM in Europe.

Therefore, although the thermodynamic predictions indicate that the aerosol system in Melpitz is most frequently sensitive to $HNO_3$, VOC controls on the conversion of $NO_x$ to $HNO_3$ suggest that $NO_x$ reductions may not be as effective in reducing inorganic aerosol concentrations, at least at first. This helps explain the results of a modeling study of the sulfate-nitrate-ammonium system over Germany that predicted $NH_3$ reductions will be more effective reducing PM than $NO_x$ reductions due to the non-linear response described above (Banzhaf et al., 2013). Such predictions are difficult to test with observations, especially because the 'accidental experiment' offered by the COVID-19 lockdown actually showed higher atmospheric $NH_3$ emissions in northwest Germany, likely due to interannual variability in meteorological parameters (Balamurugan et al., 2022).

## 4 Conclusions

Central Europe has seen dramatic changes in secondary aerosol precursor emissions, which have
decreased $NO_3^-$ and $SO_4^{2-}$ concentrations (Nec, 2024). $NH_3$ emissions have remained steady, or increased,
over the same time period, potentially changing aerosol pH, as well. In Melpitz, the annual average pH
has increased by 0.06 units per year over the 10-year period 2010 – 2019. This is similar to trends of fog
water pH in the Po Valley, Italy, where an annual increase of 0.04 units has been observed since 1993
(Paglione et al., 2021). However, inferred values of aerosol pH in the Po Valley have actually decreased
by 0.03 units per year during the same time period due to the combination of changing aerosol
composition and changing meteorology (warmer and drier). Similarly, aerosols in the southeastern US
and Canada have become more acidic despite dramatic decreases in acidic aerosol precursors, $SO_2$ and
$NO_x$ (Weber et al., 2016; Tao and Murphy, 2019), contrary to the observed trends in Melpitz. Compared
with the large body of studies on aerosol composition and trends, few studies have characterized trends
in aerosol pH and its controlling factors. These results underscore the need for more characterizations of
aerosol pH trends across Europe, and beyond.

The present results highlight the critical factors that contribute to aerosol pH in central Europe.
Similar to other locations, temperature is the most important factor driving pH variability. There is a
strong seasonal cycle to the factors affecting pH, with $NH_3$ and particle properties (composition and
concentration) also strongly affecting pH variability. The factors contributing to pH variability do not
scale with concentration or the absolute temperature or RH level. This analysis suggests that aerosol pH
will continue changing in the future with regulations and climate; however, our results suggest that
predictions of future changes in aerosol pH require a full treatment of the coupled emissions-chemistry-
meteorology system that together determine particle acidity. Although the results were derived from
measurements at a background site, similarities in other areas may also be expected. For example, in the
megacity of Paris, similarities of $NH_3$ partitioning, strong agricultural influences on secondary aerosol
formation, and $NH_4NO_3$ formation regime suggest that the observations in Melpitz may translate to urban
environments across Europe, as well (Petetin et al., 2016).

Although it is widely observed that $NH_3$ emissions in Europe peak during spring due to fertilizer
application, variability in farming practices makes this quite challenging to accurately capture in models.

For example, the region surrounding Melpitz shows large model-measurement discrepancies in $NH_3$ emissions, and predictions are worse during the spring (van der Graaf et al., 2022). This is problematic because $NH_3$ emissions in Germany contribute significantly to episodic events of elevated $PM_{2.5}$, especially in spring, which are also challenging to model (Fortems-Cheiney et al., 2016). Although $NH_3$ emissions represent a large source of uncertainty in model predictions of $NH_4NO_3$ formation, errors come from numerous sources, including $NO_x$ emissions, dry deposition rates, $HNO_3$ formation from $NO_x$ (uncertainty in OH), uncertainty in the thermodynamic partitioning, and model treatment of bidirectional flux (Petetin et al., 2016). Therefore, improved understanding of the aforementioned factors is required to advance predictive capabilities for springtime haze formation events in Europe.

Finally, these results provide additional context to the strategies that may be most effective reducing $PM_{2.5}$ mass concentrations. We applied a recently developed framework to identify the thermodynamic regimes of $NH_4NO_3$ aerosol formation through aerosol pH and ALWC (Nenes et al., 2020). Very few observations (< 1% total) were insensitive to $NH_3$ or $HNO_3$. Thermodynamic conditions indicate that $HNO_3$ was more often limiting than $NH_3$, suggesting that $PM_{2.5}$ control strategies in this region should target $NO_x$ emissions reductions, next to ammonia reductions, especially in winter. However, $NO_x$ reaction to form $HNO_3$, with subsequent partitioning to aerosol $NO_3^-$, is highly non-linear. Reductions in $NO_x$ decrease Tot-$NO_3$, but the system can respond through changes in $NO_3^-/HNO_3$ partitioning such that aerosol concentrations remain relatively stable (e.g., Womack et al., 2019). Changes in $NO_3^-/HNO_3$ partitioning can change $HNO_3$ dry deposition rates, but partitioning and dry deposition are usually large sources of model uncertainty (e.g., Zakoura and Pandis, 2019). Therefore, future studies are needed to complement the thermodynamic calculations in order to address such complexities.

**Data Availability**

All data presented in this work are available upon request.

**Author Contributions**

V.P. – formal analysis, investigation, visualization, writing – original draft, writing – review and editing. C.H. – conceptualization, supervision, formal analysis, funding acquisition, visualization, writing –

original draft, writing – review and editing. B.S. –investigation, validation, data curation, writing – review and editing. A.T. – formal analysis, investigation, data curation, writing – review and editing. L.P. - formal analysis, investigation, data curation, writing – review and editing. D.vP. - investigation, data curation, writing – review and editing. G.S. –investigation, validation, data curation, writing – review and editing. H.H. – conceptualization, supervision, resources, funding acquisition, writing – review and editing.

**Competing Interests**

The authors declare no competing financial interests.

**Acknowledgements**

The authors acknowledge support in instruments operation in Melpitz by TROPOS ACD technicians Achim Grüner and René Rabe.

**Financial Support**

C.H. and V.P. acknowledge support from the National Science Foundation through grants AGS-1719252 and AGS-2430366. The TROPOS co-authors acknowledge financial support for the deployment of the MARGA system by the German federal environment protection agency Umweltbundesamt (UBA) under contract no. 52436 as well as from the European Regional Development fund by the European Union under contract no. 100188826. Further support by the European infrastructure projects ACTRIS (EU FP7; grant no. 262254) and ACTRIS-2 (grant no. 654109) and the RI-URBANS project (grant no. 101036245) are gratefully acknowledged.

**Table 1:** Summary of annual trend analysis for species shown in Fig. 1.

| | Sulfate | Tot-NO₃[d] | ALWC[e] | Tot-NH₃[f] | NVCs[g] | pH |
|---|---|---|---|---|---|---|
| Slope[a] | -0.15 | -0.15 | -3.39 | 0.06 | 0.01 | 0.06 |
| Statistically significant[b] | Yes | Yes | Yes | No | No | Yes |
| Intercept[c] | 2.60 | 4.20 | 69.5 | 4.69 | 0.21 | 2.87 |
| $R^2$ | 0.73 | 0.80 | 0.71 | 0.05 | 0.10 | 0.75 |

[a]Linear least squares method; Units are $\mu g\ m^{-3}\ a^{-1}$, except pH: pH units $a^{-1}$

[b]Indicates whether the slope is statistically significant at the 95% confidence level

[c]Intercept relative to 2010

[d]Tot-NO$_3$ = NO$_3^-$ + HNO$_3$

[e]Aerosol liquid water content

[f]Tot-NH$_3$ = NH$_4^+$ + NH$_3$

[g]Non-volatile cations

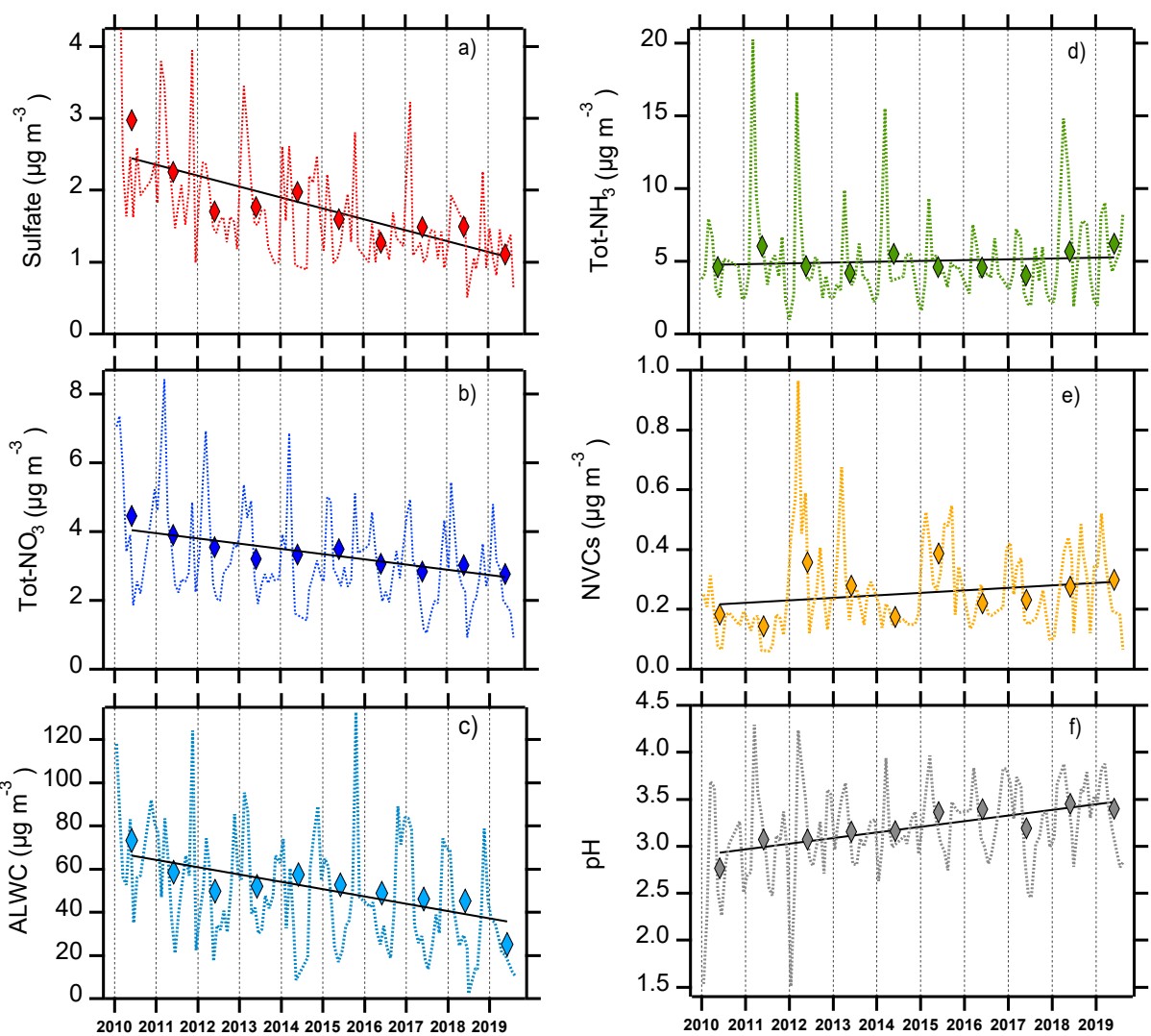

**Figure 1:** Average monthly (dotted line) and annual (diamonds) trends in a) sulfate, b) total nitrate (Tot-$NO_3$ = $HNO_3$ + $NO_3^-$), c) aerosol liquid water content (ALWC), d) total $NH_3$ (Tot-$NH_3$ = $NH_3$ + $NH_4^+$), e) non-volatile cations (NVCs), and f) aerosol pH in Melpitz over 2010 – 2019. The solid line in each panel is the linear least squares regression fit for the annual averages.

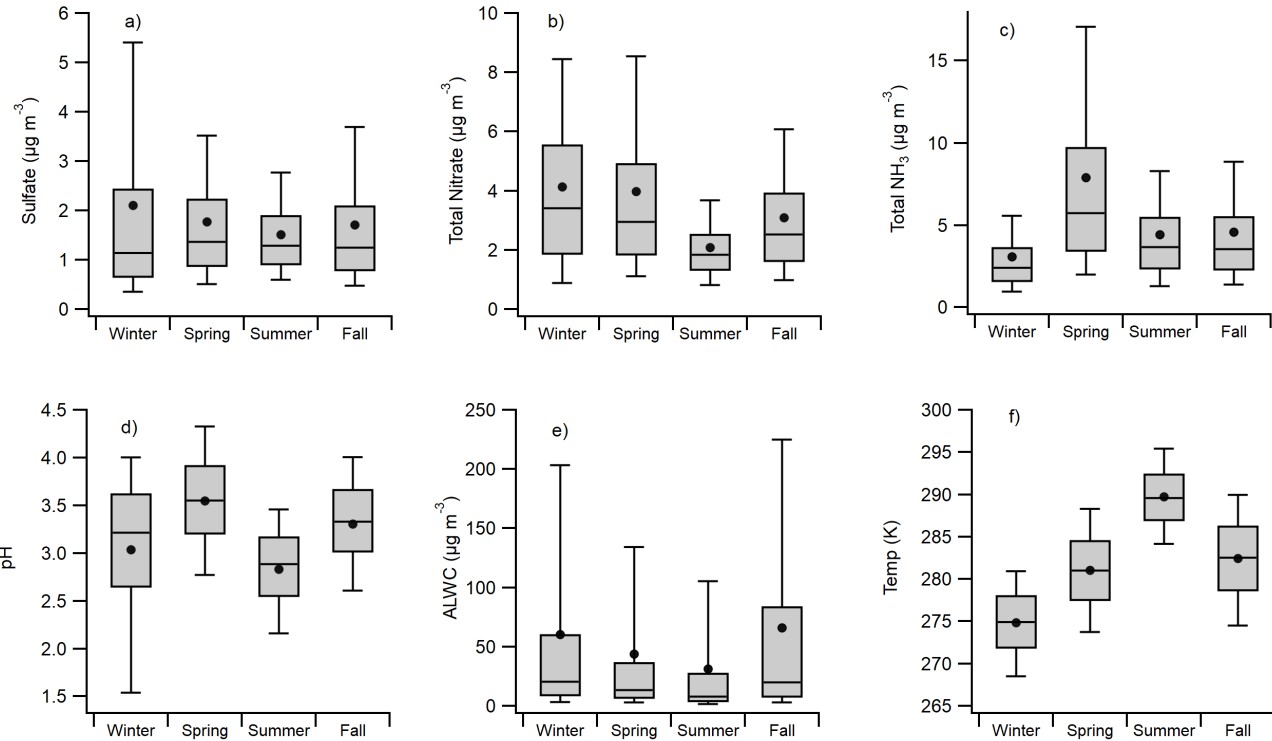

**Figure 2:** Box and whisker plots showing seasonal a) sulfate concentrations, b) total nitrate (Tot-NO$_3$ = HNO$_3$ + NO$_3^-$) concentrations, c) total NH$_3$ (Tot-NH$_3$ = NH$_3$ + NH$_4^+$) concentrations, 655 d) aerosol pH, e) aerosol liquid water content (ALWC), and temperatures. Data shown are the 10[th] and 90[th] percentiles (whiskers), quartiles (upper and lower box), median (solid line), and mean values (circles).

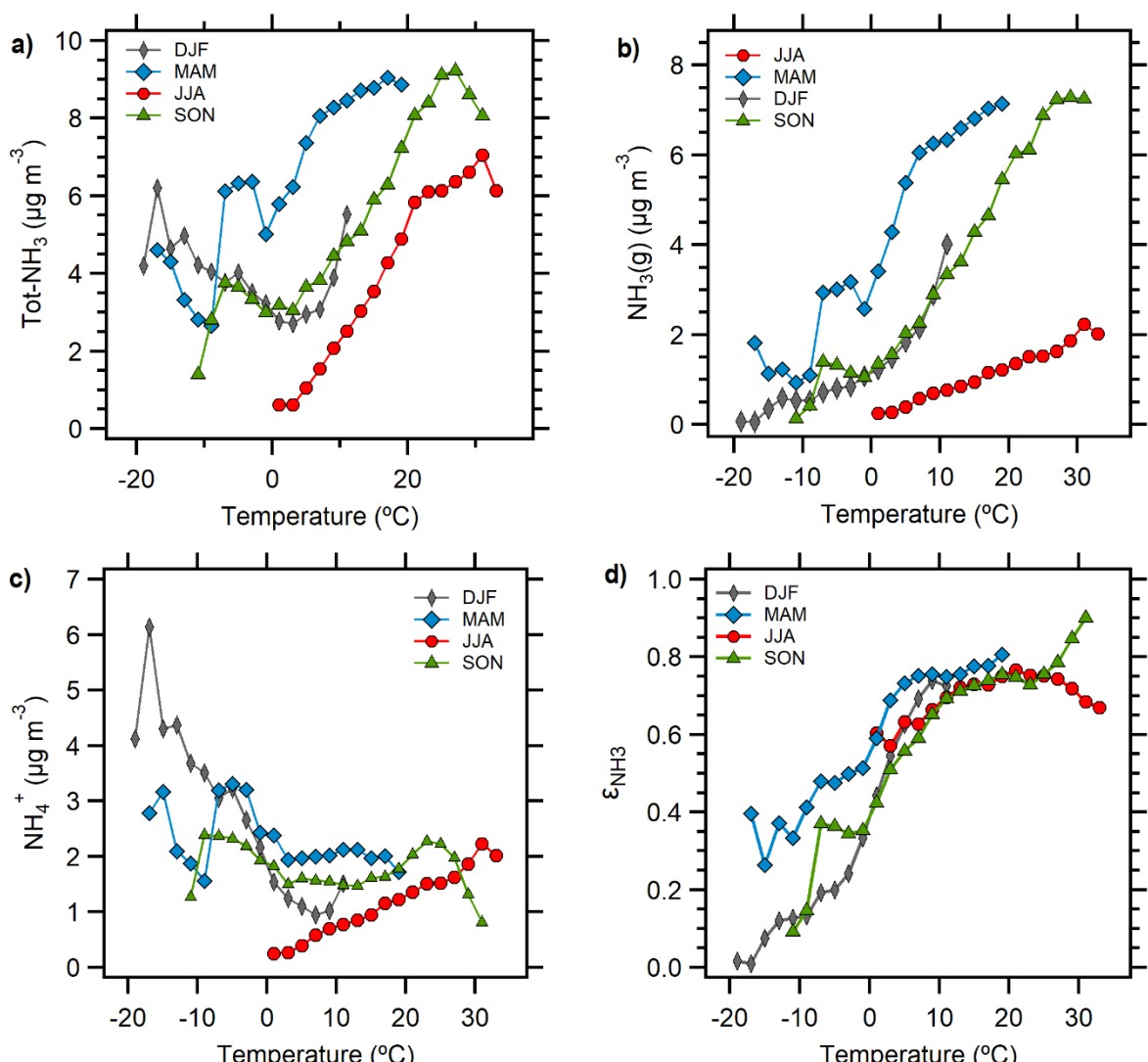

**Figure 3:** Seasonal characteristics of a) total $NH_3$ concentrations (Tot-$NH_3$ = $NH_3$ + $NH_4^+$), b) $NH_3$(g) concentrations, c) $NH_4^+$ concentrations, and $NH_3$ partitioning ($\varepsilon_{NH3}$ = $NH_3$(g)/($NH_3$(g) + $NH_4^+$)), all presented as a function of ambient temperature.

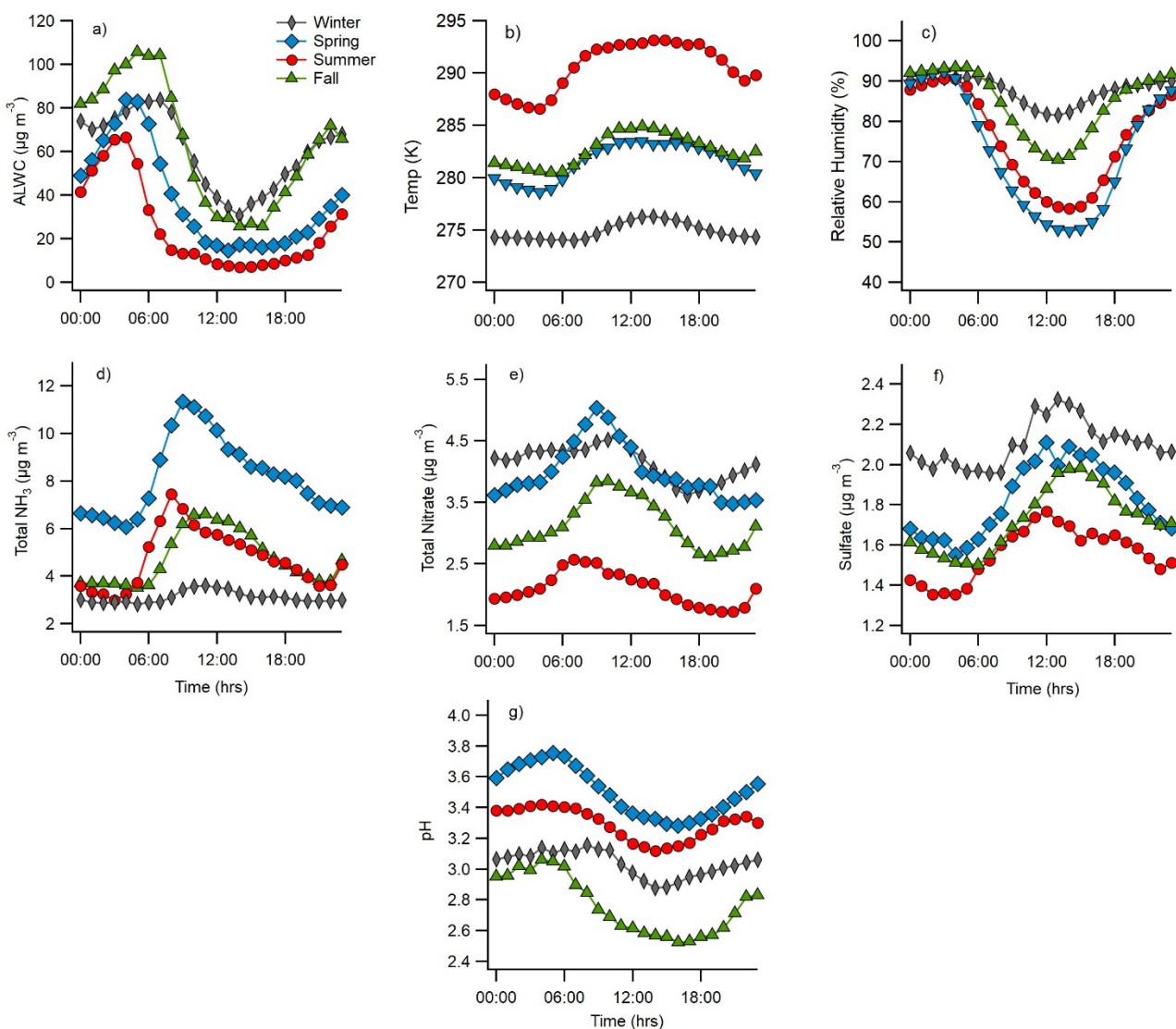

**Figure 4:** Average diurnal profiles of a) aerosol liquid water content, b) temperature, c) relative humidity, d) total $NH_3$ ($NH_3(g) + NH_4^+$), e) total nitrate ($HNO_3(g) + NO_3^-$), f) sulfate concentrations, and g) aerosol pH within each season.

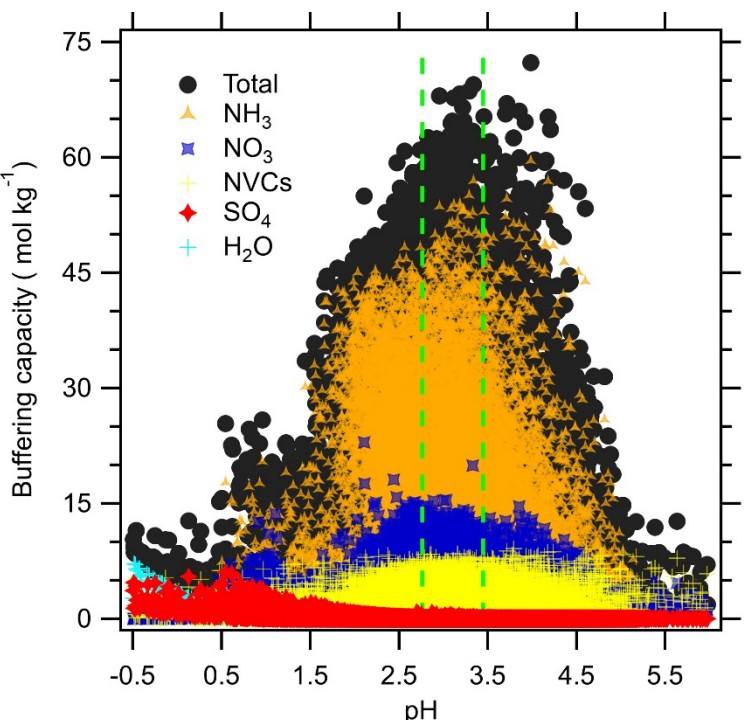

**Figure 5:** Buffering capacity calculated according to Zheng et al. (2020) evaluated for various species. Vertical green lines represent the region enclosed by mean annual pH range of this work.

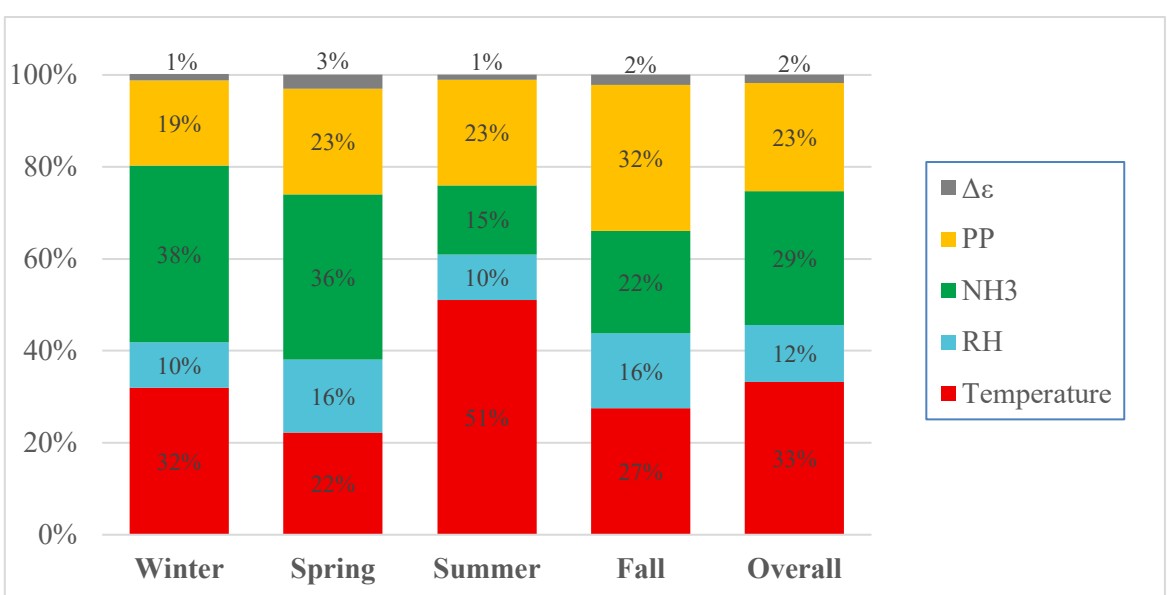

**Figure 6:** Contribution of various factors to changes in aerosol pH at Melpitz in winter, spring, summer, fall, and overall, calculated according to Tao and Murphy (2021).

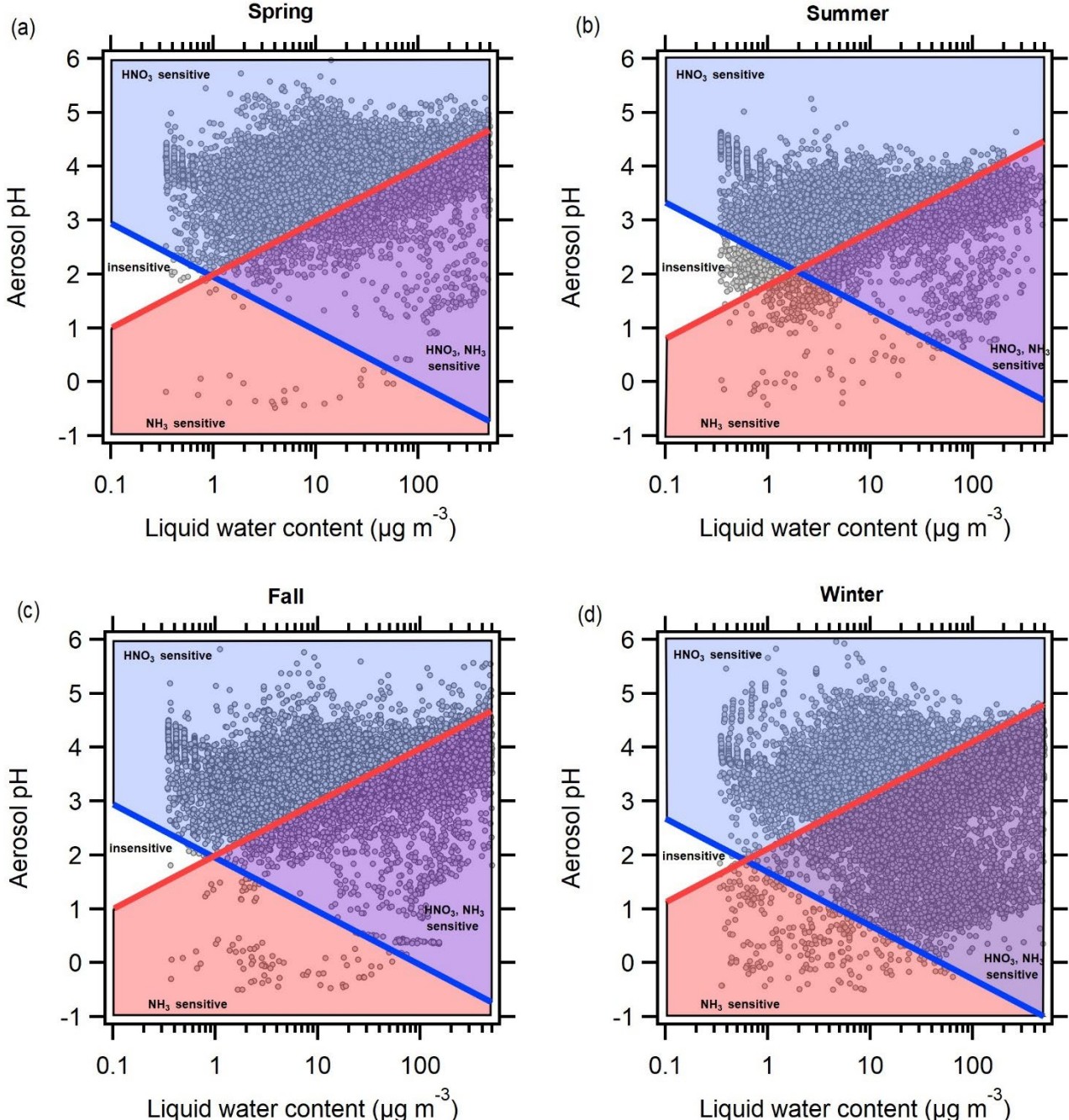

**Figure 7:** Chemical regimes in each season showing when Melpitz aerosol mass concentrations are sensitive to HNO₃ availability, NH₃ availability, both, or neither, calculated according to Nenes et al. (2020).

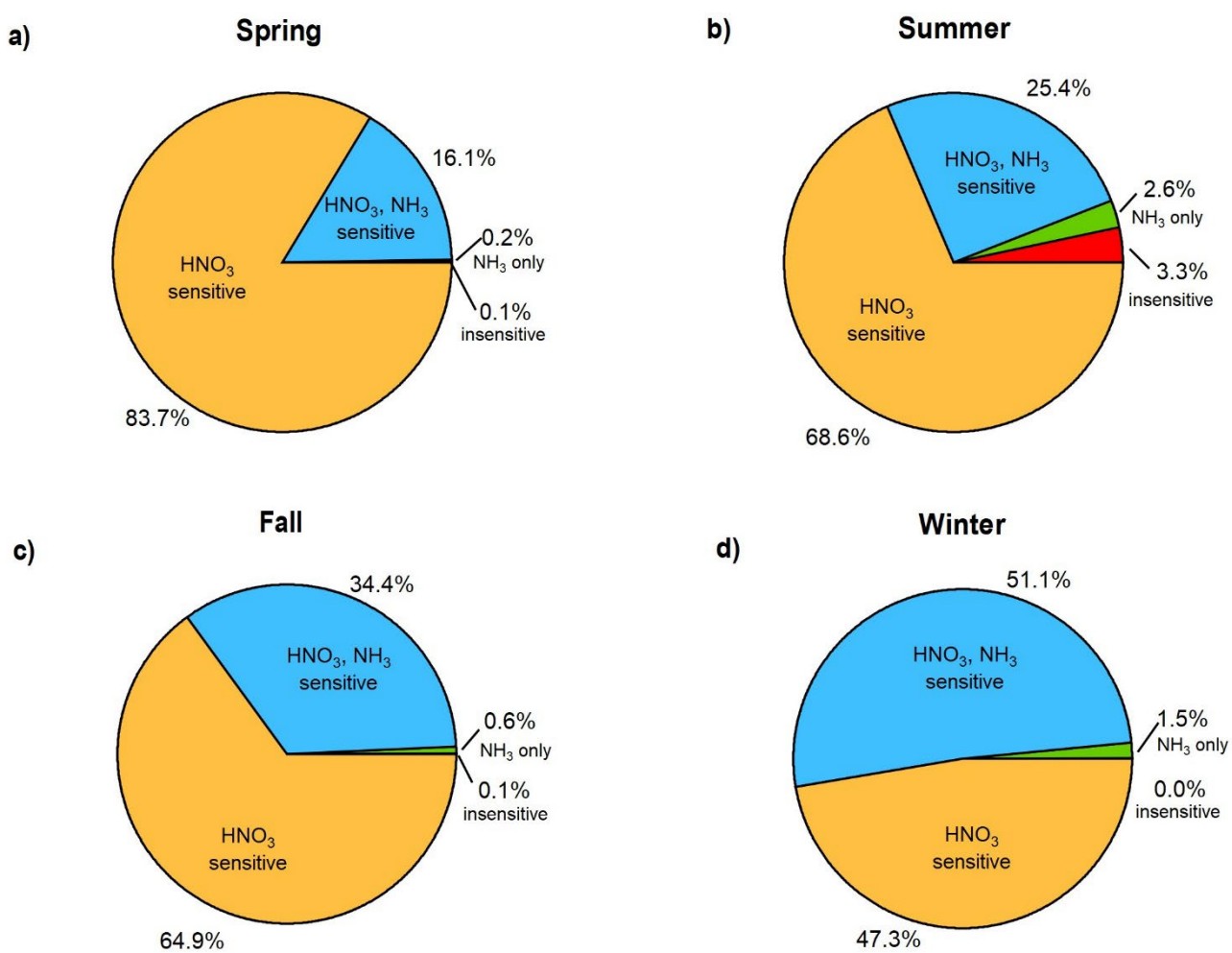

**Figure 8:** Pie charts showing the chemical regimes in Melpitz by season with the fraction of observations in which $NH_4NO_3$ aerosol mass concentrations are sensitive to $HNO_3$ availability, $NH_3$ availability, both, or neither, calculated according to Nenes et al. (2020).

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
