# Peer review of "Climatology of aerosol pH and its controlling factors at the Melpitz continental background site in central Europe"

_EGUsphere, 2025_

## Author Comment (AC1)

We thank the reviewers for their constructive comments. We have addressed each comment below, with the reviewer comment in black followed by our response in blue, which includes changes to the manuscript (highlighted in yellow).

**Referee #1**

This paper investigates the trends in aerosol liquid water content (ALWC) and pH at a rural site in Germany over a 10-year period. Factors controlling pH and an assessment on how to lower $NH_4NO_3$ concentrations through $NH_3$ or $HNO_3$ control are discussed. The data set is unique. The paper is of great interest and very well organized and easily to read. There are major issues, however. The aerosol composition data is $PM_{10}$, which greatly limits assessing fine particle pH, which is much more important than pH of $PM_{10}$, but this is never discussed. The ALWC is calculated only from inorganic species, it is not clear if OA data was available, which could be used to assess relative contribution of OA particle water. This could be important as the inorganic species concentrations decrease, as noted. The analysis is interesting, but the assessment of factors that affect pH is hard to interpret. Greater physical/mechanistic analysis would help. For example, how is one to interpret the claim of an overarching effect of T on pH variability – why does this happen? Overall, an interesting and important paper.

**Specific comments**

1. The title nor abstract state if this is $PM_{2.5}$ or $PM_{10}$, which matters greatly for pH. It is first clarified in line 105 in the methods. This is a critical issue that must be addressed up front. What is the meaning of a pH calculation that covers (mixes) fine and coarse modes that likely have highly different pHs. This is noted in lines 342 to 346 discussing changes in pH with size at this site (pH~3 for $PM_1$ and pH~4 for $PM_{1-2.5}$ ). What is the pH of Dp 2.5 to 10 um? I guess the authors are assuming that the PM chemical species included in the pH calculation are much higher in $PM_{2.5}$ or $PM_1$ than $PM_{2.5-10}$ and so pH fine particles is same as pH $PM_{10}$. This needs substantial clarification.

The reviewer raises a very important point. Kakavas et al. (2021) modeled aerosol pH over continental Europe in four size bins: $PM_1$, $PM_{1-2.5}$, $PM_{2.5-5}$, and $PM_{5-10}$. The simulations were carried out for one summer month. Across Europe, they find that smaller size bins are systematically more acidic than larger size bins due to the higher presence of non-volatile cations from sea salt and dust in larger particles, consistent with many prior studies (e.g., Keene et al., 2004; Fridlind and Jacobson, 2000; Fang et al., 2017). The average pH of $PM_1$ is 1.3 pH units lower than in $PM_{5-10}$, however, this difference exhibits a strong spatial dependence. Coastal areas with high sea salt influence, such as Denmark and northwestern France, showed the largest pH differences between different size fractions. Kakavas et al. (2021) specifically analyzed conditions in Melpitz and found the pH differences between size fractions to vary diurnally. Throughout the day, $PM_{5-10}$ has pH higher than $PM_1$ by ~1 pH unit; however, the differences between $PM_1$ and $PM_{1-2.5}$ was as large as ~1 pH unit at night but as small as ~0.25 pH units in the daytime. Very similar differences and diurnal patterns were found for the $PM_1$ vs. $PM_{2.5-5}$ fractions. Spindler et al. (2013) showed through analysis of long-term aerosol composition data that dust concentrations in Melpitz are highest during

summer. Therefore, the summertime conditions simulated by Kakavas et al. (2021) represent a maximum in the pH difference between fine and coarse particles.

In the second sentence of the abstract, we have clarified the size of particles considered here: "Here, we characterize trends in $PM_{10}$ pH and its controlling factors over the period of 2010 – 2019 at the Melpitz research station in eastern Germany, a continental background site in central Europe."

We added the following paragraph to Section 3.3 (new lines 369 – 386): "It should be noted that the present study characterizes the pH of $PM_{10}$ but small particles are systematically more acidic than large particles. Kakavas et al. (2021) modeled aerosol pH over continental Europe during summer in four size bins: $PM_1$, $PM_{1-2.5}$, $PM_{2.5-5}$, and $PM_{5-10}$. They found that smaller size bins are systematically more acidic than larger size bins due to the higher presence of non-volatile cations from sea salt and dust in larger particles, consistent with prior studies (e.g., Keene et al., 2004; Fridlind and Jacobson, 2000; Fang et al., 2017). The average pH of $PM_1$ was 1.3 pH units lower than the largest coarse bin ($PM_{5-10}$), however, this difference exhibited a strong spatial dependence. Coastal areas with high sea salt influence, such as Denmark and northwestern France, showed the largest pH differences between different size fractions. Kakavas et al. (2021) specifically analyzed conditions in Melpitz and found the pH differences between size fractions to vary diurnally. Throughout the day, $PM_{5-10}$ was ~1 pH unit higher than $PM_1$; however, the differences between $PM_1$ and $PM_{1-2.5}$ were as large as ~1 pH unit at night but as small as ~0.25 pH unit in the daytime. Very similar differences and diurnal patterns were found for the $PM_1$ vs. $PM_{2.5-5}$ fractions. We expect the simulations of Kakavas et al. (2021) to represent an upper bound on pH differences between fine and coarse fractions in Melpitz because their simulations were specific to summer: Spindler et al. (2013) showed through analysis of long-term aerosol composition data that dust concentrations in Melpitz are highest during summer. Therefore, we expect the present results to inform the trends of aerosol pH in smaller size fractions ($PM_1$ and $PM_{2.5}$), with pH values ~0.25-1 unit lower, depending on the season and time of day."

2. In the abstract it states: "Aerosol liquid water content (ALWC) decreased by 50% during the analyzed time period in response to decreasing sulfate and nitrate. Aerosol pH exhibited an increase of 0.06 units per year, a trend that was distinct from other regions". Be consistent? I.e., give each change per year or overall change during the measurement period or both. Stating that this is a change in pH of 0.6 units in 10 years (line 219), or give a % change would provide a direct comparison.

We modified the sentence so that it now reads: "Aerosol liquid water content (ALWC) associated with inorganic species decreased by 3.4 µg m$^{-3}$ a$^{-1}$, which resulted in a 50% decrease during the analyzed time period in response to decreasing sulfate and nitrate." We do not apply the same convention (i.e., x% increase in pH) to the pH discussion because of the logarithmic pH scale and the very different meaning of a pH of 0 vs. a concentration of 0.

3. Also, the 50% decrease in ALWC does not include contribution of organic species, this is a calculated quantity, and it should be clearly noted this number is based only inorganic species – a major, and possibly very impactful limitation since it may not be the actual change in ALWC, especially since inorganic species are decreasing.

   Yes, thank you for pointing out that important detail. We have updated the text so it now reads: "Aerosol liquid water content (ALWC) associated with inorganic species decreased by 50% during the analyzed time period in response to decreasing sulfate and nitrate." See also our response to the comment below where we discuss in more detail the issue of ALWC associated with organic compounds.

4. Finally, how does ALWC associated with $PM_{10}$ compare to PM fine, i.e., how to interpret this data.

   We emphasize that our analysis is on $PM_{10}$ – we do not have $PM_1$ data so we cannot do a direct comparison between the two. In the study by Kakavas et al. (2021), discussed in detail above in our response to Comment #1, they also modeled the ALWC associated with different aerosol size fractions across Europe. $PM_1$ had – by far – the highest ALWC concentrations in continental areas. ALWC associated with $PM_{1-2.5}$ was also high in coastal areas with high sea salt concentrations. ALWC associated with $PM_{2.5-5}$ and $PM_{5-10}$ was consistently low, due to the high concentrations of dust in these fractions. We have added the following sentence to clarify this point: "Although the model output represents ALWC associated with $PM_{10}$, Kakavas et al. (2021) showed that ALWC was many times higher in fine particles than coarse mode particles across Europe."

5. In the abstract: "Seasonal analysis showed strong variability in factors controlling aerosol pH…." Is this referring to variability between seasons or within a season. Does the term *variability overall* mean for the complete measurement period?

   We have clarified this sentence so that it now reads: "The factors controlling aerosol pH varied by season."

6. Last part of Abstract regarding NOx controls to lower PM $NH_4NO_3$, why NOx control, what about factors that affect OH or $O_3$, ie VOCs, since oxidants may have more control over $HNO_3$ than NOx). A number of modeling papers have looked into this. Isn't it more correct to call it factors that control $HNO_3$?

   The reviewer is correct that it is more accurate to discuss factors that control $HNO_3$. We have clarified this point in the abstract: "Thermodynamic analysis of the aerosol system shows that secondary inorganic aerosol formation is most frequently $HNO_3$ limited, suggesting that factors that control $NO_x$ would be more effective than $NH_3$ controls in reducing PM mass concentrations. However, the non-linear response of gas-phase $HNO_3$ and aerosol $NO_3^-$ to $NO_x$ emissions in the region, likely due to VOC controls on oxidant formation and subsequent impacts on $NO_x$ conversion to $HNO_3$, highlights the challenge associated with PM reductions needed to attain new air quality standards in this region."

We also modified the discussion in Section 3.5 so that it now reads: "A similar phenomenon was also observed in the Salt Lake Valley, Utah, where wintertime ammonium nitrate formation did not respond to $NO_x$ controls, but was instead sensitive to VOC emissions (Womack et al., 2019). This occurs because $NO_x$ conversion to $HNO_3$ is oxidant-limited in some environments. In such cases, control of VOC emissions, rather than $NO_x$ or $NH_3$, can provide the most direct effect on aerosol $NH_4NO_3$ (Womack et al., 2019; Dang et al., 2023). Changes in $NO_3^-/HNO_3$ partitioning can also reduce the $HNO_3$ dry deposition sink, increasing the atmospheric lifetime of Tot-$NO_3$ (Zhai et al., 2021). This complex chemistry is consistent with a recent study that found that a 23% reduction in $NO_x$ emissions during the COVID-19 lockdown period was associated with only a ~5% decrease in $PM_{2.5}$ mass in Germany (Balamurugan et al., 2022). Further, actual emissions control measures need to account for costs and analysis by Liu et al. (2023) indicates $NH_3$ controls are 5-10 times more cost effective than $NO_x$ controls at reducing PM in Europe.

Therefore, although the thermodynamic predictions indicate that the aerosol system in Melpitz is most frequently sensitive to $HNO_3$, VOC controls on the conversion of $NO_x$ to $HNO_3$ suggest that $NO_x$ reductions may not be as effective in reducing inorganic aerosol concentrations, at least at first."

7. Line 55, the role of T is critical, (as noted many times further on in the paper), I suggest this be expanded more with a few sentences. Might what to look at DOI: 10.1126/sciadv.ado4373 (or https://www.science.org/doi/10.1126/sciadv.ado4373) Eg, if most pH buffering is by $NH_3/NH_4^+$, $NH_3$ volatility (and water vapor volatility) is very important. More on this below regarding how to interpret the importance of T.

We thank the reviewer for pointing out this reference, which we agree supports our discussion. We have added this reference in the Introduction. See our response to Comment #11 below for more discussion on the role of temperature in affecting pH.

8. Line 70. See the following recent paper for more pH estimated over years. https://doi.org/10.1038/s41561-024-01455-9

We thank the reviewer for highlighting this reference – it has been added to the manuscript.

9. Line 145 and on. $NH_3/NH_4+$ partitioning measured vs compared looks good, as is practically always the case in other studies. What about $HNO_3/NO_3^-$, $HCl/Cl^-$, which often does not look so good? If $HNO_3/NO_3^-$ is not so good, what are the implications for the findings of this paper, such as, does it affect predictions on $HNO_3$ vs $NH_3$ control?

We agree with the reviewer that the partitioning of other species can be instructive to examine in regard to model evaluation. In practice, this was challenging for the Melpitz dataset because the gas-phase $HNO_3$ measurements by the MARGA may have had a

systematic bias, discussed in detail by Stieger et al. (2018). Comparison of the gas-phase $NH_3$ measurements by the MARGA indicated much closer agreement with other independent measurements. While the MARGA measured both HCl and aerosol $Cl^-$, the concentrations of HCl were on average ~0.1 – 0.2 µg m$^{-3}$, so we chose to use the more abundant $NH_3/NH_4^+$ pair for this analysis.

10. In Equation 1, what is the variable on the bottom left in within the bracket, $X(NH_4^+)$, mole faction?

    Yes, this is the mole fraction of ammonium – the description has been added to the text.

11. It would be helpful (one could even say critical) to give a physical explanation for all the factors that are derived in Eq 3. Example, how does the $NH_3$ factor alter pH, how does the T factor alter pH? Please do this for all factors so that the results can be interpreted later in the paper. Are the underlying mechanism affecting these factors mutually exclusive, eg, does T have an effect on many of them? If so, what does the T factor mean and how is one to interpret the results of this analysis? Overall, I find this approach very confusing, possibly overly simplistic, due to lack of mechanistic explanations.

    The reviewer raises an excellent point here (and similar in Comments #7, #15, #16, #17, and #18). Reviewer 2 raised a similar concern. Because of the similarities, we address all of these comments here.

    The method of Tao and Murphy (2021) that we adopt here derives the pH sensitivity (i.e., their Equation 8 or our Equation 2) based on the partitioning of $NH_3/NH_4^+$. Essentially, they quantify the impact of the various individual factors on $NH_3/NH_4^+$ phase partitioning and then translate this quantitatively to an effect on aerosol pH. The method relies on the assumption that a factor that affects equilibrium $NH_3/NH_4^+$ phase partitioning also affects aerosol pH; an extensive body of literature supports this assumption (notably the work of Nenes and Weber; the Pye et al. (2020) review article on acidity; as well as the Zheng et al. (2020) multiphase buffer theory paper).

    The framework of Tao and Murphy (2021) assesses the following factors for their influence on $NH_3/NH_4^+$ phase partitioning, and hence pH: (1) temperature, (2) gas-phase $NH_3$ concentration, (3) RH, (4) particle properties, and (5) an error term associated with simplifying assumptions in the derivation. Temperature affects equilibrium constants ($K_H$, $K_a$, and $K_w$) that affect $NH_3/NH_4^+$ phase partitioning. The $NH_3$ (g) abundance clearly affects $NH_3$ partitioning, as a $NH_3/NH_4^+$ system at equilibrium will shift towards the gas phase in response to a reduction in $NH_3$ and towards the particle phase in response to an $NH_3$ increase. RH directly affects ALWC, which impacts the particle-phase activities of aerosol species, including $NH_4^+$ and $H^+$, both of which directly factor into the computation of $NH_3/NH_4^+$ phase partitioning. Finally, particle properties account for the particle hygroscopicity, which also influences ALWC, and the ammonium activity coefficient, which affects the ammonium activity. We have added the following text to Section 2.3 to clarify this point: "The framework of Tao and Murphy (2021) assesses the

following factors for their influence on $NH_3/NH_4^+$ phase partitioning, and hence pH: (1) temperature, (2) gas-phase $NH_3$ concentration, (3) RH, (4) particle properties, and (5) an error term associated with simplifying assumptions in the derivation. Temperature affects equilibrium constants ($K_H$, $K_a$, and $K_w$) that affect $NH_3/NH_4^+$ phase partitioning. Duan et al. (2025) demonstrated that T effects on activity coefficients and semi-volatile vapor pressures also contribute to its overarching influence on pH. The $NH_3$ (g) abundance clearly affects $NH_3$ partitioning, as a $NH_3/NH_4^+$ system at equilibrium will shift towards the gas phase in response to a reduction in $NH_3$ and towards the particle phase in response to an $NH_3$ increase. RH directly affects ALWC, which impacts the particle-phase activities of aerosol species, including $NH_4^+$ and $H^+$, both of which directly factor into the computation of $NH_3/NH_4^+$ phase partitioning. Finally, particle properties account for the particle hygroscopicity, which also influences ALWC, and the ammonium activity coefficient, which affects the ammonium activity."

The reviewer is correct that the various factors are connected in the atmosphere. For example, T and RH are often inversely correlated (on diurnal timescales), while $NH_3$ emissions respond strongly to temperature (see a similar comment from Referee #2). Importantly, these factors often act on pH in opposite directions. For example, increased T lowers aerosol pH through changes in the thermodynamic equilibrium constants, while increased $NH_3$(g) increases aerosol pH by shifting more $NH_3$ to the condensed phase. Tao and Murphy (2021) quantify these connections through their 'square of standard deviation' analysis. Performing such an analysis on our dataset is beyond the scope of our study, however, we point the reviewer to our discussion in Section 3.4 that we believe provides clarity on this topic: "It should be noted that that the various factors contributing to pH variability are not completely independent. For example, temperature and $NH_3$ often influence pH in opposite directions, but $NH_3$ emissions and abundance have a clear dependence on temperature (Fig. 3). Similarly, temperature and RH are typically inversely related at ground level. RH affects ALWC, which changes solute concentrations in aqueous particles and gas-particle partitioning of semi-volatile compounds (Ansari and Pandis, 1999). Temperature is a master variable that affects most processes in the environment – with regards to aerosol pH, it affects compound vapor pressures, equilibrium constants, solubilities (i.e., overall partitioning), and reaction rates (Pye et al., 2020; Tilgner et al., 2021). Therefore, while their effects on aerosol pH can be separately quantified, these factors can somewhat offset each other in the ambient atmosphere relative to conditions where one factor changes independently of the others. A similar phenomenon was observed and noted for the Canadian data set analyzed by Tao and Murphy (2021)."

12. Lines 207 to 210 states (SNA) are the most abundant inorganic aerosol components in Melpitz (Spindler et al., 2013), and thus, they control ALWC abundance. True, if only inorganic species are included in the calculation of ALWC. Consider OA water, as noted before.

The is an excellent point. We clarified in the abstract that we are only considering ALWC associated with inorganic components (Comment 3 above). We have also added the following discussion to the Methods section where we discuss the modeling approach: "The E-AIM model was run without organics as inputs, a good assumption that has minimal effect on the predicted pH under the imposed RH limitation (Battaglia et al., 2019). Organic aerosol components can take up water and contribute substantially to ALWC, so the results presented herein do not account for the fraction of ALWC associated with those organic species. Based on AMS measurements of $PM_1$ composition at the Melpitz research station, it is likely that water content associated with organics was ~20% of the total ALWC assuming an average kappa value of 0.15 for the organics (Atabakhsh et al., 2025), though this would change with the aerosol composition."

13. Line 230 to 236. Please explain how (mechanistically why) pH of aerosol particles and cloud/fog water trends would be related in an environment of changing PM chemical species concentrations. As one example, does $NH_3/NH_4^+$ buffering apply to cloud/fogs as it does to fine PM (or I guess $PM_{10}$ in this case)?

The reviewer raises an interesting question about buffering in clouds/fogs compared to aerosol systems. We do not have the data needed to perform a detailed analysis of buffering in cloud and fog water. Indeed, in the Po Valley study, the increasing trend in fog water pH was accompanied by a predicted decrease in the aerosol pH, so these are not necessarily linked. Investigating this question would likely represent an entire paper, itself, and thus is beyond the scope of our current analysis.

14. Lines 347 re Fig 5 and buffering capacity. If Melpitz has a higher buffer capacity that explains dampened diurnal pH variability why does the same site have a larger trend in pH interannually (ie, the trend over the 10 years of the study)? Can the authors give a physical explanation of why $NH_3/NH_4^+$ has the highest buffering capacity, i.e., higher than other chemical species. Could it be that the gas-particle partitioning for $NH_3/NH_4^+$ is closer to 50:50 compared to the other semi-volatile components. In fact, maybe this explains why the pH is in the predicted range.

The primary factor that contributes to why $NH_3/NH_4^+$ has the highest buffering capacity is the overall abundance of total $NH_3$ ($NH_3 + NH_4^+$) concentrations compared to others (e.g., compare Fig. 4d and 4e, noting the different y-axis scales). In Equation 4, the total molality-based concentration of the buffering acid/base pair is part of the calculation of buffering capacity. Note that Fig. 4 presents mass concentrations, so the relative difference of the molal concentrations of Tot-$NH_3$ vs. Tot-$NO_3$ (or other species) will be even greater (i.e., the total molal concentration of Tot-$NH_3$ is significantly higher). The reviewer is correct that the pH conditions frequently lead to high concentrations of $NH_3$ in both the gas- and particle phases, with median epsilon-$NH_3$ (epsilon-$NH_3$ = $NH_3(g)$/Tot-$NH_3$) values that range from 0.47 (winter) to 0.86 (summer). This also contributes to the buffering capacity of $NH_3/NH_4^+$.

15. Regarding section 3.4 Drivers of pH variability. This section goes into detail on what factors can be attributed to most of the model-predicted pH variability. It is noted that T

is the largest overall driver since it has effects in multiple ways. Why not interpret all the trends by looking at the effect of T in each season and see if it provides a more unifying explanation. For example, how much of what is observed is because of T effect on volatility of the semi-volatile species ($NH_3$, $HNO_3$, $H_2O$). When T is lower, more $NH_3$ partitions to the particle due to less volatile, same for $H_2O$, both may move the $NH_3/NH_4+$ to more or less buffering (need to know $NH_4$ epsilon), and is that consistent with observed greater $NH_3$ buffering in winter. Looking at S curves one may get a physical sense of what is going on. This is essentially stated in lines 426 to 428 at the start of section 3.5 (although could add $H_2O$ to this list). Unless some in-depth explanations are provided, the method of Tao and Murphy may obscures insight.

A new article was published in April 2025 that enhances our discussion of this point (Duan et al., 2025). We now cite this article, and we included it in our discussion in Sections 2.3 and 3.4. See also our response to Comment #11 above for additional discussion of the importance of T in affecting pH.

16. Line 373 to 374. Explain this line, lower T produced increase in pH, higher T a lower pH. Why does that happen?

See our response to Comment #11 above.

17. As noted in line 387, the factors are not independent, even within the thermodynamics of the process (eg, excluding other external factors such as effect of T on ambient $NH_3$ concentrations). Take the stated variables, T and $NH_3$ concentration. Mechanistically, how does T and $NH_3$ concentration change pH. (This question was asked above as part of the section where the equations are presented). Are they related, eg, T affects volatility of $NH_3$, lower T results in more $NH_3$ to $NH_4^+$ which raises pH which then affects other things that go back and influence pH and NH3 concentration. Eg, T affects the epsilon ($NH_3$), which for a constant total ammonia ($NH_3 + NH_4^+$) leads to change in $NH_3$ concentration. So, what is the meaning of the T effect vs the $NH_3$ concentration effect? Is it even meaningful to try and separate these two effects?

See our response to Comment #11 above.

18. Can a physical explanation for line 401 to 402 be provided, which states: During winter, $NH_3$ was a more important driver of pH variability than temperature (38% and 32%, respectively), but temperature was far more important during summer (51% to 15%). If one does not know what the T and $NH_3$ factors are influenced by, this is hard to interpret.

See our response to Comment #11 above.

19. Again, questions about OA water. Section 3.5, specifically discussing how ALWC, eg lines 450 to 455, what if organic species, such as WSOC, significantly affect ALWC, which is not considered here. What is the implication? This is another useful aspect of the Nenes formulation since you could make similar plots with ALWC that includes OA. Or

since OA water does not greatly affect pH, add OA ALWC with appropriate Kappa values and see how this shifts the data relative to the regimes. Or run ISORROPIA Lite.

Unfortunately, we do not have the data required to compute the OA ALWC for each observation. We have added the following discussion to Section 3.5 to clarify this point: "In the present analysis, we only considered ALWC associated with inorganic aerosol components. While water-soluble OA likely exerts minor effects on pH (Battaglia et al., 2019), it can contribute to the ALWC. Consideration of the ALWC contributed by inorganic and organic compounds would result in a minor shift of the individual data points in Fig. 7, although the logarithmic scale indicates very few points would shift to a different thermodynamic regime (e.g., from insensitive to $HNO_3$-sensitive).".

20. As noted before, support the assertion that NOx is the key emission driving $HNO_3$ concentration, lines 460.

    See our response to Comment #6 above.

21. Comment regarding Figure 7. The location of the lines (blue and red) delineating the regimes appear exactly the same in all four seasons, (all four plots), despite substantial temperature differences (ie, line 434-435: 275 K in winter, 281 K in spring, 291 K in summer, and 282 K in fall). Is that correct?

    Yes, the reviewer is correct that this was a mistake. The updated figure has been added to the manuscript (and is also shown below). As described in Section 3.5, the analysis was done with the temperature-sensitive values used and the results in Fig. 8 reflect the seasonal differences in temperature.

[Figure]

**Referee #2**

The authors investigate diurnal, seasonal, and annual trends of aerosol pH, liquid water content, inorganic species, and their controlling factors at a continental background site in Germany over a 10-year period. They found that temperature and ammonia was the most important factors in controlling the aerosol pH, and that $NH_4NO_3$ formation was most frequently $HNO_3$ limited, suggesting that NOx controls would be more effective that $NH_3$ controls in reducing particulate mass concentrations. The findings are scientifically sound, the manuscript is very well written and interesting. I recommend publishing it after the authors address the following comments.

**Comments**

1. In section 2.2, it is mentioned that HCl and Cl- were measured, but these species were not discussed or shown in any Figures. Was chloride always below detection limit? Please clarify.

   The primary reason $Cl^-$/HCl were not discussed in the manuscript is that Stieger et al. (2018) dedicated extensive analysis and discussion to the sources and atmospheric processes influencing these concentrations. They found the dominant source of $Cl^-$ to be transport of marine aerosol from the North Sea while the dominant source of HCl is acid displacement. In addition, the concentrations of HCl and $Cl^-$ were significantly lower than sulfate, nitrate, and ammonium (and their gaseous analogs). Stieger et al. (2018) report Tot-Cl (Tot-Cl = HCl + $Cl^-$) concentrations were typically in the 0.2 – 0.4 µg m$^{-3}$ range, and both species were below detection limit for about 50% of all samples.

2. While the seasonal and annual trends of total inorganic species can be readily understood as shown in Figure 1, the roles of temperature and relative humidity on gas/particle partitioning of semivolatile species (HNO$_3$ and NH$_3$) and pH are a bit unclear. Temperature directly modulates the Henry's Law constant and other equilibrium constants that govern the partitioning of HNO$_3$, HCl, and NH$_3$ as well as bisulfate ion dissociation. At the same time, temperature together with absolute humidity govern the relative humidity, which in turn governs the equilibrium aerosol liquid water content. Thus, the dependence of pH and PM mass on temperature and RH is rather complex and intertwined. For instance, pH and ALWC show inverse diurnal behavior with respect to temperature, but diurnal variation of RH would also have a strong influence on ALWC and thus also on pH. I suggest adding a panel in Figure 4 for the diurnal profiles of RH for each season to show a more complete picture of the dependence of pH and ALWC on T and RH.

   Referee #1 made a similar comment - please see our detailed reply to Comment #11 above. We agree with the Referee's suggestion to update Figure 4. The new version of Fig. 4 with the seasonal diurnal RH profiles added is below.

[Figure]

**References**

Atabakhsh, S., Poulain, L., Bigi, A., Coen, M. C., Pöhlker, M., and Herrmann, H.: Trends of PM₁ aerosol chemical composition, carbonaceous aerosol, and source over the last 10 years at Melpitz (Germany), Atmos. Environ., 346, 121075, doi:10.1016/j.atmosenv.2025.121075, 2025.

Duan, X., Zheng, G., Chen, C., Zhang, Q., and He, K.: Driving factors of aerosol acidity: a new hierarchical quantitative analysis framework and its application in Changzhou, China, Atmos. Chem. Phys., 25, 3919–3928, https://doi.org/10.5194/acp-25-3919-2025, 2025.

Stieger, B., Spindler, G., Fahlbusch, B., Müller, K., Grüner, A., Poulain, L., Thöni, L., Seitler, E., Wallasch, M., and Herrmann, H.: Measurements of PM₁₀ ions and trace gases with the online

system MARGA at the research station Melpitz in Germany - A five-year study, Journal of Atmospheric Chemistry, 75, 33-70, 10.1007/s10874-017-9361-0, 2018.

*Note: other references can be found in the revised manuscript.